# Subglacial precipitates record Antarctic ice sheet response to late Pleistocene millennial climate cycles

Gavin Piccione [1] ✉, Terrence Blackburn [1], Slawek Tulaczyk [1], E. Troy Rasbury [2], Mathis P. Hain [1], Daniel E. Ibarra [3,4], Katharina Methner [5], Chloe Tinglof[1], Brandon Cheney[1], Paul Northrup [2] & Kathy Licht[6]

Ice cores and offshore sedimentary records demonstrate enhanced ice loss along Antarctic coastal margins during millennial-scale warm intervals within the last glacial termination. However, the distal location and short temporal coverage of these records leads to uncertainty in both the spatial footprint of ice loss, and whether millennial-scale ice response occurs outside of glacial terminations. Here we present a >100kyr archive of periodic transitions in subglacial precipitate mineralogy that are synchronous with Late Pleistocene millennial-scale climate cycles. Geochemical and geochronologic data provide evidence for opal formation during cold periods via cryoconcentration of subglacial brine, and calcite formation during warm periods through the addition of subglacial meltwater originating from the ice sheet interior. These freeze-flush cycles represent cyclic changes in subglacial hydrologic-connectivity driven by ice sheet velocity fluctuations. Our findings imply that oscillating Southern Ocean temperatures drive a dynamic response in the Antarctic ice sheet on millennial timescales, regardless of the background climate state.

One of the persistent challenges involved in both reconstructions and projections of global mean sea level is determining what sectors of the Antarctic Ice Sheet (AIS) are vulnerable to significant retreat, the timescales of such retreat, and the conditions that trigger ice loss events[1]. Modern observations[2,3] of retreating ice near marine-terminating ice sheet margins demonstrate the potential for rapid AIS mass fluctuations brought on by changing Southern Ocean temperature[4] (hereafter referred to as ocean-thermal forcing). The key link between this ocean-thermal forcing and ice sheet mass lies in the delivery of heat to the ice sheet margins, which affect ice shelves and grounding lines. Ice sheet stability is regulated by ice shelves[5] and grounding line positions[6], which are vulnerable to thinning and retreat

when contacted by warm ocean waters. Ice sheet models suggest that ice shelf decay can result in enhanced flow of grounded ice up to 1000 km upstream of the grounding lines of large outlet glaciers and ice streams[7]. On millennial timescales this feedback could cause substantial velocity changes in these fast-flowing ice drainage pathways[8], ultimately affecting continent-wide ice sheet mass balance[9].

Millennial-scale Southern Ocean temperature oscillations are driven by a feedback between ocean-atmosphere teleconnections that is modulated by Atlantic Meridional Overturn Circulation (AMOC):[10] the mean state ocean circulation responsible for cross-equatorial heat transport from the Southern Hemisphere to the Northern Hemisphere. Changes to the intensity of AMOC result in out-of-phase polar

[1]Earth and Planetary Sciences, University of California Santa Cruz, Santa Cruz, CA, USA. [2]Department of Geosciences, Stony Brook University, Stony Brook, NY, USA. [3]Department of Earth and Planetary Science, University of California Berkeley, Berkeley, CA, USA. [4]Department of Earth, Environmental and Planetary Science, and the Institute at Brown for Environment and Society, Brown University, Providence, RI, USA. [5]Department of Geological Sciences, Stanford University, Stanford, CA, USA. [6]Department of Earth Sciences, Indiana University-Purdue University Indianapolis, Indianapolis, IN, USA. ✉e-mail: gpiccion@ucsc.edu

temperature cycles[11] recorded by isotopic climate proxies in ice cores, identified as Dansgaard-Oeschger cycles in the Greenland ice core records and Antarctic Isotope Maxima (AIM) events in Antarctic ice core records. This oceanic teleconnection, called the bipolar seesaw, also affects atmospheric circulation by regulating the temperature gradient between the middle and high latitudes[12], which shifts the intertropical convergence zone north when AMOC rate is high (Northern Hemisphere/Southern Hemisphere warm/cold periods) and south when AMOC rate is decreased (Northern Hemisphere/ Southern Hemisphere cold/warm periods)[13]. As the intertropical convergence zone migrates southwards during AIM events, Southern Hemisphere westerly winds experience parallel latitudinal shifts and strengthening[14], causing upwelling of relatively warm circumpolar deep waters onto the Antarctic continental shelf[15]. Antarctic marginal ice is affected by these upwelling cycles, which deliver circumpolar deep waters to the base of ice shelves and grounding lines, triggering enhanced basal melting and retreat during Southern Hemisphere millennial warm periods[4].

Although ice sheet models[9] and modern observations[2,3] indicate that the AIS is susceptible to ice loss through ocean-thermal forcing, regional differences in ice bed topography, drainage geometry, and ice thickness[16] in peripheral sectors of Antarctica may lead to geographic differences in grounding line vulnerability, adding spatiotemporal complexity to ice sheet response. Millennial-scale climate oscillations also vary in intensity depending on the background climate state, where large continental ice sheets during glacial periods[17] and enhanced atmospheric $CO_2$ concentrations during full interglacial conditions[18,19] dampen the amplitude of millennial-scale climate variability. In contrast, intermediate climate states are characterized by more frequent, larger magnitude changes in polar temperature[20]. Therefore, geologic evidence of AIS evolution across a wide geographic range and diverse climate states is necessary to support simulations of suborbital changes in ice mass. However, existing geologic records documenting millennial-scale AIS mass loss[21–23] are limited to bipolar-seesaw events during the last two glacial terminations, are constrained by low-resolution age models, and are restricted spatially to the Weddell Sea Embayment and offshore sediments. This leaves the regional extent and magnitude of AIS response to suborbital climate change unconstrained.

Here, we present observations from an archive of subglacial hydrologic evolution recorded by chemical precipitates that formed >900 km apart beneath the East Antarctic Ice Sheet (EAIS), over a combined >100 kyr period during the Late Pleistocene. This dataset provides a sequence of high-resolution U-series age constraints of ice sheet evolution in response to millennial-scale climate change. Mineralogic and geochemical variations in subglacial precipitates provide evidence for periodic changes in subglacial hydrologic connectivity between the AIS interior and margin that occur contemporaneously with bipolar seesaw-related Southern Hemisphere climate cycles. Combining precipitate data with a reduced-complexity model of ice sheet thermodynamics, we demonstrate a link between subglacial hydrologic conditions and millennial-scale changes in ice sheet velocity.

## Results

### Changes in subglacial precipitate mineralogy correlated with millennial climate cycles

In this study, we report geochronological and geochemical results collected from two subglacial precipitates that formed over tens of thousands of years in subglacial aqueous systems on the EAIS side of the Transantarctic Mountains (TAM) (Fig. 1a). Sample MA113 comes from Mount Achernar Moraine (henceforth MAM; 84.2°S, 161°E), a nearly motionless body of blue ice on the side of Law Glacier[24], The moraine is located ca. 20 km downstream of the polar plateau, with debris derived locally from Beacon Supergroup and the Ferrar Group[25]. Sample PRR50489 was found at Elephant Moraine (henceforth EM; 76.3°S, 157.3°E), a supraglacial moraine in a blue ice area of Transantarctic Mountains, where -100 ka of ice sublimation has released debris from basal ice of the East Antarctic ice sheet[26,27], which also consists predominantly of rocks from Beacon and Ferrar[28]. Precipitates form in subglacial water, and are transported to the ice surface within upward-flowing sections of glacier ice before being exposed on the surface as the surrounding ice sublimates[25]. The exhumation of precipitates within basal ice sections makes it difficult to precisely locate their formation site. However, constraints on ice velocities and sublimation rates can be used to place their most likely formation area within ca. 10 km of where they were collected. For example, the length of time for the emergence of basal debris to MAM is estimated to be at least 35 ka[25]. Considering the youngest radiometric U-series age obtained for sample MA113 (25.44 ka), there was little time for the precipitate to travel over horizontal distances, supporting a formation area proximal to the MAM. A similar emergence time of ca. 40–50 ka can be estimated for PRR50489, but a greater minimum formation time (147 Ka) allows the possibility of a longer horizontal distance traveled. However, if we assume a basal transport velocity scaled similarly to other EAIS basal ice[24], PRR50489 could have traveled only 10 km of horizontal distance in 100 kyr. Hence, we infer that both samples MA113 and PRR50489 likely formed in basal overdeepenings found within several kilometers upstream of EM and MAM[16], which offer suitable settings for precipitate formation because they would allow subglacial water bodies to persist over long time periods (Supplementary Note 1).

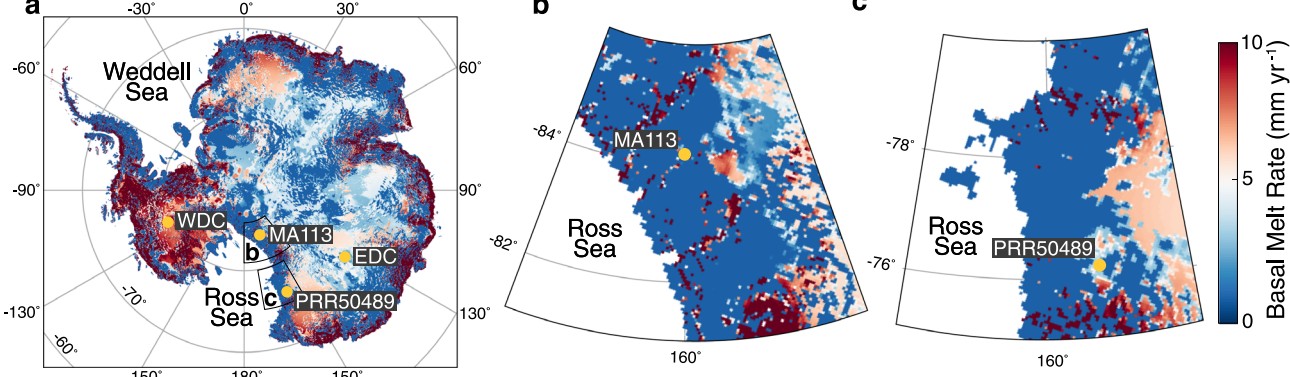

**Fig. 1 | Antarctic mean basal melt rate. a** Map of estimated modern mean basal melt rate[67] truncated at 10 mm yr⁻¹. **b** Map zoomed in to show basal melt rate near location of MA113. **c** Map zoomed in to show basal melt rate near location of PRR50489. Data sourced from ref. 67. Data licensed under: https://creativecommons.org/licenses/by/3.0/.

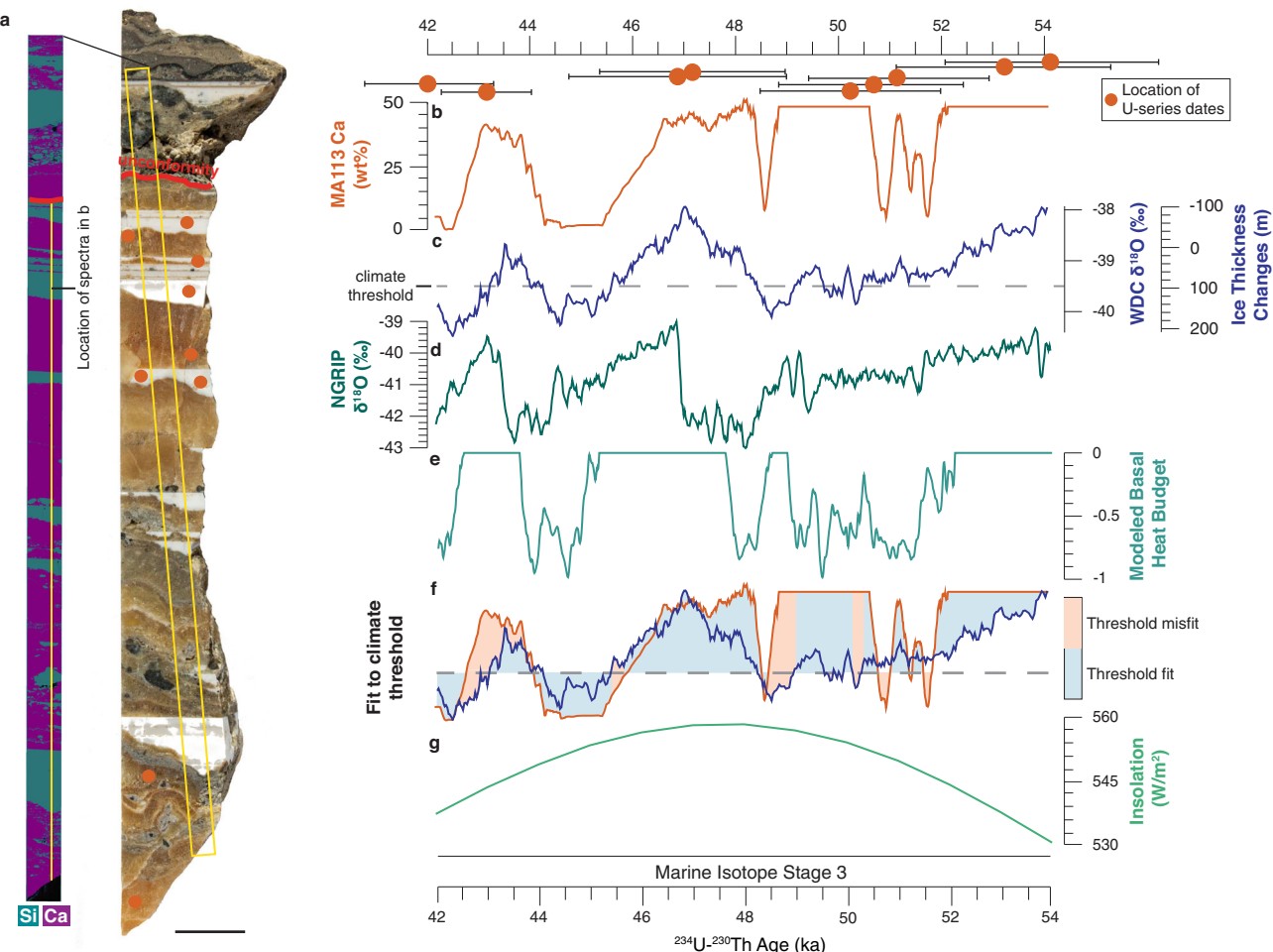

**Fig. 2 | Sample MA113 SEM-EDS image and comparison to climate records.**
**a** Slab and SEM-EDS image of sample MA113. Scale bar is 1 cm. **b** Ca concentration of subglacial precipitate sample MA113. High values represent calcite precipitation; low values represent opal precipitation. U-series dates include 2σ uncertainties bars. **c** δ¹⁸O measured in West Antarctic Divide Ice Core (WDC)[107]. Gray dashed line delineates threshold value for magnitude of ice thickness change necessary to elicit subglacial hydrologic response. **d** δ¹⁸O measured in Northern Greenland Ice Core Project (NGRIP)[108]. **e** Reduced complexity model of ice sheet thermodynamics (RCMIST) output of basal heat budget over the formation timeframe of sample MA113 in units of mm/year of equivalent basal freezing rate. Negative values indicate freezing. Positive values correspond to basal melting and are truncated at

0 mm/yr. Forcing for RCMIST is provided by ice thickness changes at the foothills of the Transantarctic Mountains, which are parameterized as a linear function of the ice core isotopic record. The magnitude and scale of these thickness changes is shown on the y-axis in **c**. **f** Binary measure of fit between mineral cyclicity in **b** and climate threshold on WDC data in **c**. A fit is defined as points where both WDC values fall above the climate threshold and calcite is precipitated, or WDC values fall below the threshold and opal is precipitates. Otherwise, the point is considered misfit. **g** Southern Hemisphere summer insolation (75°S) over the period of MA113 formation. The record in **d** is synchronized to AICC2012 chronology; the record in **b** is synchronized WD2014 chronology. Isotope ratios are on the VSMOW (Vienna Standard Mean Ocean Water) scale.

Samples PRR50489 and MA113 are 3 and 9 cm thick respectively, with alternating layers of calcite and opal-A (Figs. 2a, 3a, b) implying cyclic changes in the subglacial environment. Unlike subglacial precipitates forming in alpine settings[29,30] and beneath the Laurentide ice sheet[31], PRR50489 and MA113 do not display characteristics indicative of formation by regelation or in a basal film, and instead require cm-scale or deeper subglacial cavities that remain open on >10 kyr time-scales. Textures within each of these samples indicate that opal and calcite form via two different mechanisms. Calcite layers nucleate on the substrate and form acicular (MA113; Fig. 2a) or bladed (PRR50489; Fig. 3a, b) crystals in botryoidal shapes. Opals fill void space between calcite crystals and form distinct layers with flat tops, implying formation from nucleation in the water column followed by particle settling. While diagenetic transformation in these samples cannot be completely ruled out, petrographic analyses of calcite layers show no evidence for calcite dissolution or reprecipitation (Supplementary Fig. 1 and 2), and X-ray diffraction data from opal suggest that they are present in the opal-A form (Supplementary Fig. 3; Supplementary Note 4).

We measured ²³⁴U-²³⁰Th ages on eleven opal layers from PRR50489 that constrain the timeframe of precipitation from 230 to 147 ka (Fig. 2b), and ten opal and calcite layers from MA113 ranging in age from 55 to 42 ka (Fig. 3c). We construct a stratigraphic age model for each sample using a Bayesian Markov chain Monte Carlo model in which the principal of superposition is imposed on each dated layer to refine age estimates based on stratigraphic order[32] (Supplementary Fig. 4). Depth profiles of Si and Ca concentration collected using Energy Dispersive X-ray Spectroscopy provide a continuous representation of sample mineralogy: with high Ca areas representative of calcite and low Ca areas representative of opal. We pair stratigraphic age models with Ca concentration spectra to create timeseries describing the oscillations of precipitate mineralogy (Figs. 2b, 3c). These mineralogic timeseries reveal a temporal cyclicity in opal deposition, with opal layers in PRR50489 precipitated every 8–10 kyr between marine isotope stages 7 and 6, and opal layers in MA113 precipitated every 2–4 kyr during marine isotope stage 3. To investigate a possible link between cycles of precipitate mineralogy and climate, we compare timeseries for each precipitate with climate proxies

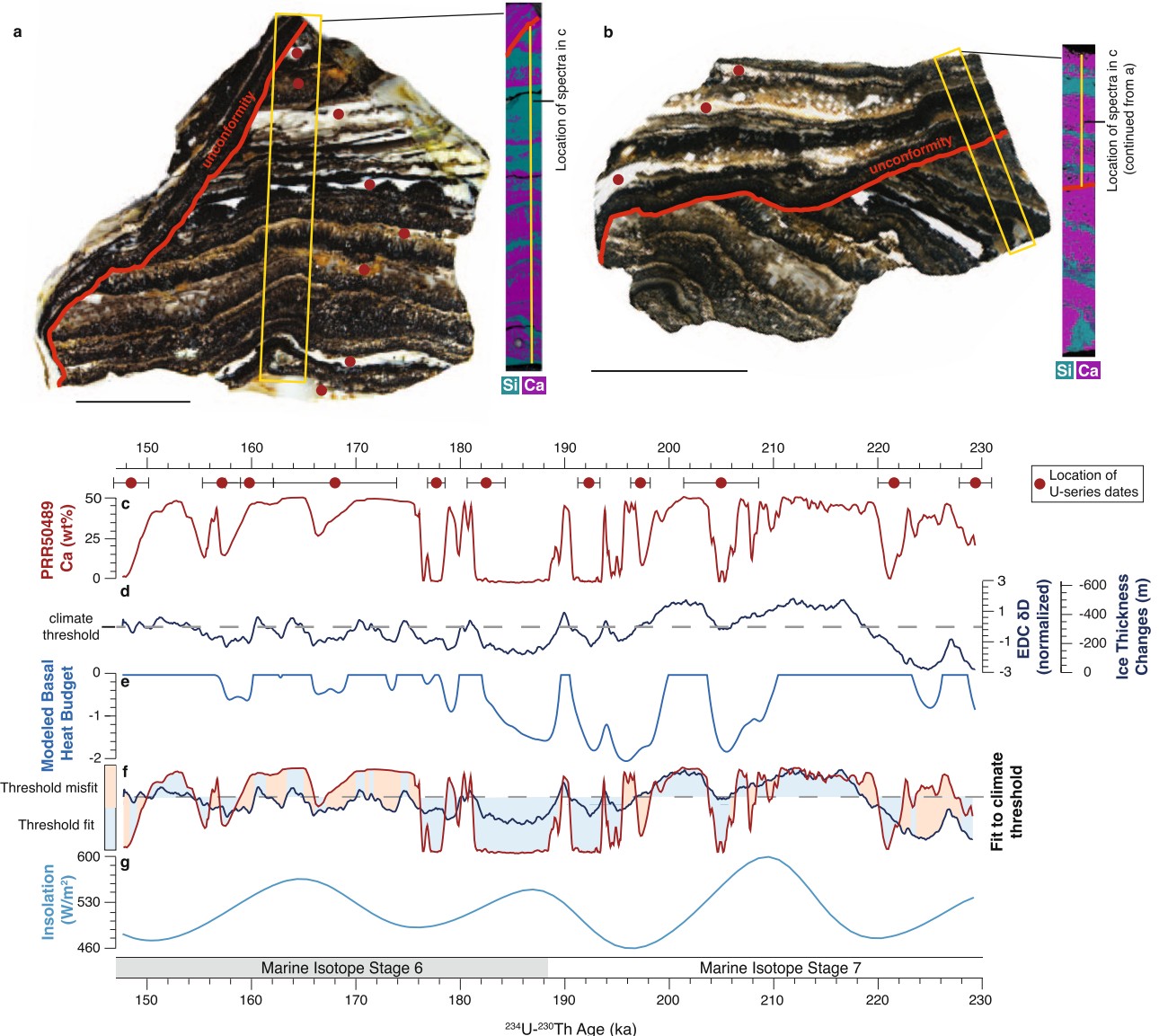

**Fig. 3 | Sample PRR50489 SEM-EDS image and comparison to climate records.**
**a** Slab and SEM-EDS image of sample PRR50489. **b** Slab and SEM-EDS image of
second piece of sample PRR50489. This piece of sample includes material
above angular unconformity. Scale bars are 1 cm. **c** Ca concentration of sub-
glacial precipitate sample PRR50489. High values represent calcite precipita-
tion; low values represent opal precipitation. U-series dates include 2σ
uncertainties bars. **d** δD measured in EPICA Dome C Ice Core (EDC)[109,110]. EDC
record is detrended and converted to a z-score by zero-mean normalization to
eliminate orbital trends. Gray dashed line delineates threshold value for mag-
nitude of ice thickness change necessary to elicit subglacial hydrologic
response. **e** Reduced complexity model of ice sheet thermodynamics (RCMIST)
output of basal heat budget over the formation period of sample PRR50489 in
units of mm/year of equivalent basal freezing rate. Negative values indicate
freezing. Positive values correspond to basal melting and are truncated at
0 mm/yr. Forcing for RCMIST is provided by ice thickness changes at the
foothills of the Transantarctic Mountains, which are parameterized as a linear
function of the ice core isotopic record. The magnitude and scale of these
thickness changes is shown on the y-axis in **c**. **f** Binary measure of fit between
mineral cyclicity in **c** and climate threshold on EDC data in **d**. A fit is defined as
points where both EDC values fall above the climate threshold and calcite is
precipitated, or EDC values fall below the threshold and opal is precipitates.
Otherwise, the point is considered misfit. **g** Southern Hemisphere summer
insolation (75°S) over the period of PRR50489 formation. Record in **d** is syn-
chronized to AICC2012 chronology.

in both Antarctic (Figs. 2c, 3d) and Greenland (Fig. 3d) ice cores. Visual
comparison between Ca-spectra and Antarctic temperature proxies
reveals a consistent, linear relationship between climate cycles and
precipitate mineralogy, with calcite formation (high Ca wt%) during
warm AIM peaks, and opal formation (low Ca wt%) during Antarctic
cold periods (Figs. 2, 3). Yet this visual comparison does not consider
that climate forcing is a continuum, while precipitate mineralogy is
binary. As such, there exists for each sample a threshold in climate
forcing (Figs. 2c, 3d: horizontal bar) above/below which precipitate
mineralogy is predominantly calcite/opal. The apparent linear syn-
chrony between precipitate mineralogy and Southern Hemisphere

temperature indicates that bipolar-seesaw-driven climate change
triggers variability in EAIS subglacial environments. A more quantita-
tive comparison between the sample mineralogy and climate records
that considers a threshold response requires additional discussion of
the glaciologic processes controlling this relationship and is presented
in a later section.

**Millennial-scale cycles in subglacial hydrologic connectivity**
To understand the link between ocean-atmosphere-cryosphere tele-
connections and the mineralogy of subglacial precipitates, we first uti-
lize geochemical and isotopic measurements to characterize the

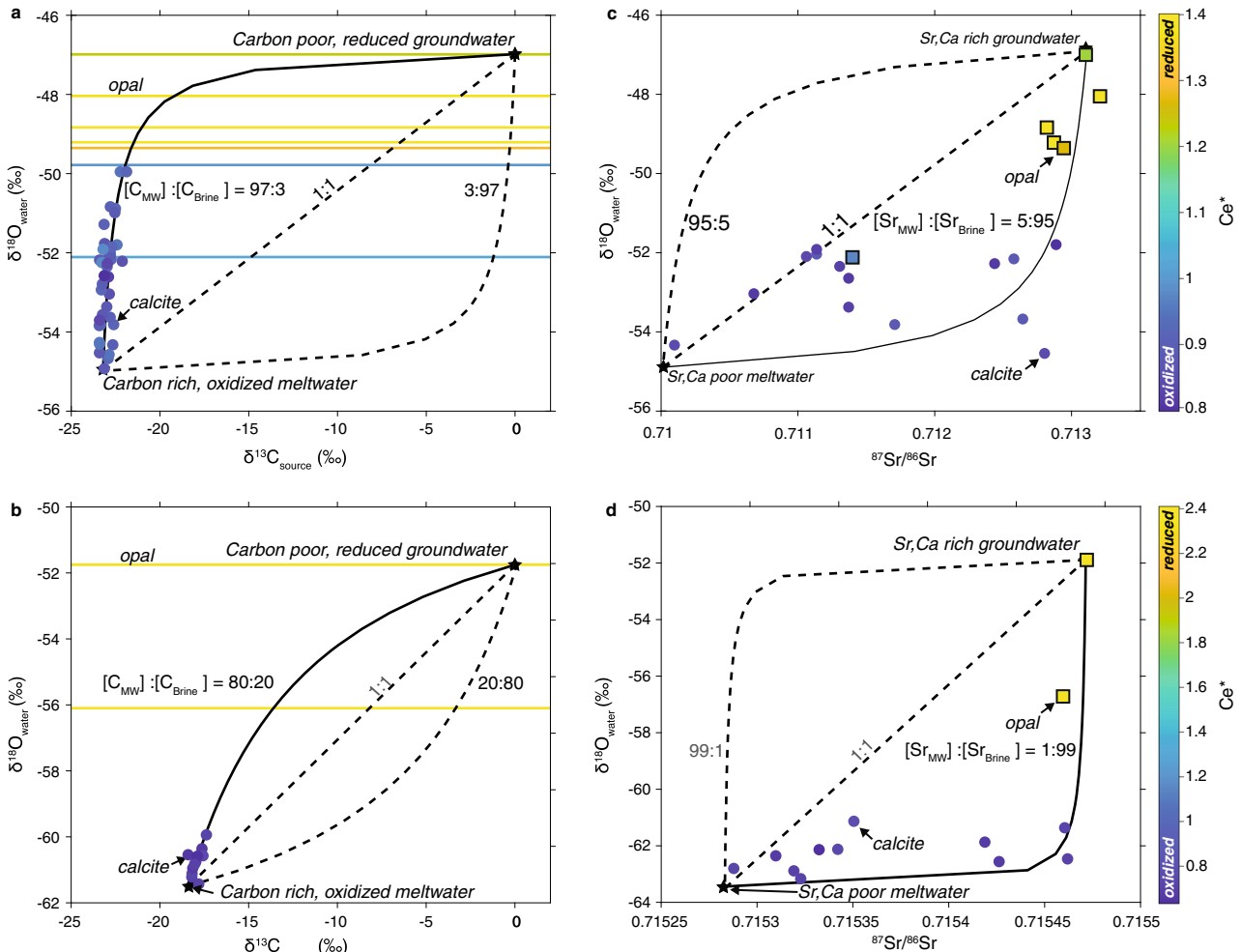

**Fig. 4 | Stable isotope mixing models for precipitates PRR50489 and MA113.**
**a** $\delta^{18}O$ of the precipitating fluid versus $\delta^{13}C_{CaCO3}$ (we interpret $\delta^{13}C_{CaCO3}$ to indicate the composition of the C source), plotted for PRR50489 calcite and opal. Endmembers (stars) include an opal precipitating fluid that is a carbon poor, brine with lower oxygen contents; and a calcite precipitating fluid that is a carbon rich, oxidizing meltwater. Solid curved line represents mixing model between the two endmembers[111]. To fit calcite data, mixing model requires meltwater to have a total carbon concentration 40-fold higher than brine. **b** As in **a**, but data are from sample MA113. To fit calcite data, mixing model requires meltwater to have a total carbon concentration fivefold higher than brine. **c** $\delta^{18}O$ of the precipitating fluid versus $^{87}Sr/^{86}Sr$ composition of the Sr source, plotted for PRR50489 calcite (circles) and opal (squares). To fit data, mixing model requires meltwater to have a total strontium concentration 20-fold lower than brine. **d** As is **c**, but data are from MA113. To fit data, mixing model requires meltwater to have a total strontium concentration 50-fold lower than brine. Dashed lines represent mixing models with different C or Sr ratios. All data are color coded by Ce* value, with blue being the lowest, most oxidizing values, and yellow being the highest, most reducing values. Oxygen isotopic compositions corrected to water compositions assuming equilibrium fractionation during calcite formation and a formation temperature of 0 °C.

precipitate source fluids. The carbon ($\delta^{13}C_{VPDB}$) and oxygen ($\delta^{18}O_{VSMOW}$) isotopic compositions of opal and calcite-forming waters are distinct for both PRR50489 (Fig. 4a) and MA113 (Fig. 4b), with calcites forming from waters with low $\delta^{18}O$ values, and opals forming from waters with $\delta^{18}O$ values up to 7‰ higher. The low $\delta^{18}O$ compositions of the calcite endmember suggest origination from meltwaters generated beneath the EAIS interior[33,34], likely in conjunction with modest additional $^{18}O$-depletion occurring as meltwaters experience freezing in transit to the ice sheet margin[35]. The heaviest $\delta^{18}O$ compositions of the opal endmember fluid (−46.15‰ for PRR50489 and −52.10‰ for MA113) are similar to the $\delta^{18}O$ of ice proximal to the region where samples were exhumed[34,36], suggesting that these waters originate as basal meltwater formed closer to the ice sheet margin. Another distinguishing characteristic of opal and calcite-forming waters are their cerium anomalies (Ce*), a proxy for redox conditions[37] (Figs. 4, 5c). In both samples, Ce* correlates with sample mineralogy, with calcite Ce* values indicating precipitation from oxidizing waters (Ce*<1), while the most $^{18}O$-enriched opals exhibit Ce* values indicating precipitation from intermediate to reduced waters (Ce*>1) (Figs. 4, 5c). In most cases, $\delta^{18}O$ values of opal

scale with Ce*, pointing to variable mixing ratios between an oxidizing and a more reducing water during the formation of both minerals (Fig. 4a, b). The carbon isotopic composition of calcite ($\delta^{13}C_{CaCO3}$) from both samples are $^{13}C$-depleted (−23‰ for PRR50489 and −18‰ for MA113) suggesting that carbon is sourced from similarly $^{13}C$-depleted subglacial organic matter (~−26‰ in PRR50489) that is oxidized during microbial respiration. Similarly $^{13}C$-depleted carbon is observed in other EAIS basal aqueous systems[38,39]. For water closed off from the atmosphere, microbial respiration can function as the only significant source of $CO_2$. This $CO_2$ undergoes hydrolysis to $H_2CO_3$ and is utilized in the chemical weathering of the substrate. In an area with bedrock dominated by silicate materials, most or all of the $HCO_3^-$ in the system will result from silicate weathering reactions, which would result in $\delta^{13}C$ of dissolved inorganic carbon within 3‰ of $\delta^{13}C$ of the source carbon[40,41]. The -5‰ offset between $\delta^{13}C_{CaCO3}$ of PRR50489 and MA113 can be caused either by a slightly different organic source or contact with bedrock that contains carbonate.

Calcite layers in both samples exhibit trends in $\delta^{13}C$ and $\delta^{18}O$ compositional space that suggest mixing between two isotopically

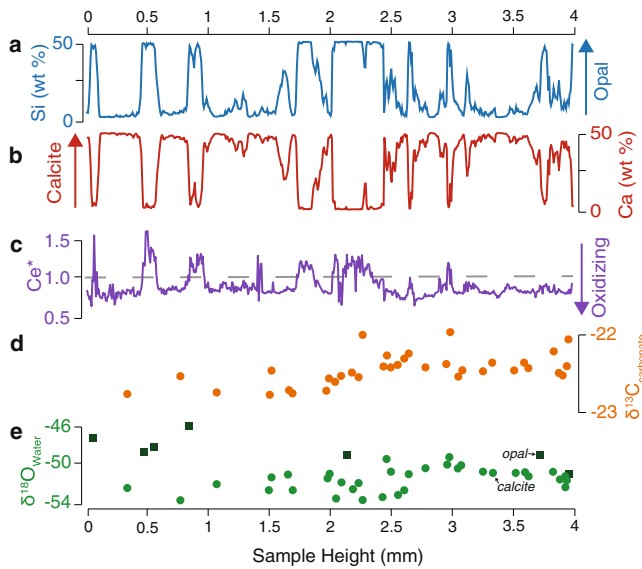

**Fig. 5 | Geochemical Data from PRR50489. a** LA ICP-MS Si concentration curve. **b** LA ICP-MS Ca concentration curve. **c** LA ICP-MS Ce* curve. **d** $\delta^{13}C$ data from calcite layers. **e** $\delta^{18}O$ data from calcite and opal layers. Opal layers are represented by areas with high Si and low Ca concentrations; calcite layers have high Ca concentration and low Si concentration.

distinct fluids with different solute concentrations (Fig. 5). To match the trends in the calcite data for PRR50489 and MA113, a $^{13}C$- and $^{18}O$-depleted, calcite-forming endmember water must have 40-fold and 5-fold higher total carbon concentration respectively, relative to a low-carbon, opal-forming endmember water. Though $\delta^{13}C$ of the opal-forming water cannot be directly measured, for the mixing curve to fit calcite compositions and opal $\delta^{18}O$ values, the opal-forming endmember waters must have higher $\delta^{13}C$ values ($\delta^{13}C > -5$ ‰): a composition comparable to that of sub-AIS brines[42]. Similar mixing relationships are observed between the $^{87}Sr/^{86}Sr$ and $\delta^{18}O$ composition of opals and calcite (Fig. 4c, d), requiring endmember waters to be distinct in both Sr concentration and isotopic composition. In both samples, the opal-forming waters have more radiogenic (higher) $^{87}Sr/^{86}Sr$ and higher $\delta^{18}O$ than the calcite-forming waters. Mixing between strontium and oxygen show that 20- to 50-fold of the total Sr in the system originates in the opal-forming endmember (Fig. 4c, d). Due to their similar geochemical behavior, strontium and calcium concentrations in saline waters scale proportionally[43]. On this basis, the two endmember fluids must have distinctive Ca concentrations, with the opal-forming endmember accounting for >95% of Ca in the system. This mixing relationship affirms the prevalence of two endmember waters with divergent concentrations: a highly Ca-rich, C-poor opal-forming brine that dominates the aqueous cation budget, and a relatively Ca-dilute, C-rich calcite-forming meltwater that adds oxygen and carbon to the system.

Together, redox and isotopic data permit the identification of suitable analogs for both endmember waters. The opal endmember is characterized by low carbon and high calcium concentrations, a $^{13}C$-enriched $\delta^{13}C$ composition, a $\delta^{18}O$ composition that matches ice proximal to the TAM, and a Ce* value indicative of intermediate to reducing fluids. Collectively, these data support opal formation from a subglacial brine with limited oxygen. A potential analog matching these criteria are $CaCl_2$ brines that emanate from beneath the modern EAIS in the McMurdo Dry Valleys (MDV)[44]. In addition to the aforementioned similarities between the opal endmember and MDV brines, $\delta^{234}U_o$ and bulk rare earth element compositions of MDV brines[45] (Supplementary Fig. 5) match that of the precipitate opals. In comparison, the calcite endmember water has high carbon and low calcium

concentrations, low $\delta^{13}C$ and $\delta^{18}O$ compositions, and a Ce* signature indicative of oxidizing fluids. Combined, these geochemical signatures support a water composition analogous to glacial meltwater originating beneath the interior domes of East Antarctica, which would form from waters with highly $^{18}O$-depleted, oxygen-rich meltwater from dome-ice, can have high concentrations of $^{13}C$-depleted carbon from microbial respiration[38,46], and would be much more dilute than marginal brines. One analog for this glacial meltwater endmember is C-rich, low salinity jökulhlaup water measured at Casey Station[47] that flushed from subglacial lakes beneath Law Dome, and resulted in subglacial aragonite precipitation during an AIM warm period[48].

To test if water mixing is a plausible mechanism for the observed opal and calcite layers, we use the frezchem database[44] within the geochemical program PHREEQC[49] to simulate mixing of the two endmember waters identified in the above-mentioned geochemical analyses (see Methods for full description of PHREEQC models). Monomineralic opal layers represent periods of amorphous Si saturation, which can occur in subglacial environments through cryoconcentration[50]. Supplementary Fig. 6 shows a set of PHREEQC simulations that demonstrate opal saturation during freezing of a $CaCl_2$ brine, where, unlike the result predicted from any other surface waters, the deficiency of carbon in the brine precludes calcite precipitation. Since geochemical data suggest calcite precipitation from an oxidized, carbon-rich, and isotopically distinct glacial meltwater, we explore conditions under which calcite saturates upon mixing of EAIS basal meltwater with $CaCl_2$ brine. Mixing the Casey Station jökulhlaup water with opal-forming brines, we identify a strong supersaturation in calcite over a broad range of mixing proportions (Supplementary Fig. 7), and a cessation of opal precipitation consistent with discrete calcite pulses during mixing. While we have explored alternative formation mechanisms in Supplementary Note 2, our preferred interpretation of the combined geochemical, and modeling results is that freeze-flush cycles in sub-EAIS drainage system drive the alternating opal-calcite precipitation at the base of the ice sheet.

## Millennial-scale ice sheet variability

The key finding from our subglacial precipitate archive is that millennial-scale ocean-atmosphere-cryosphere teleconnections trigger geochemical and hydrologic responses beneath the EAIS, where AIM warm phases drive enhanced delivery of interior subglacial meltwaters to the ice sheet margin. Subsequent millennial cold phases promote upstream expansion of basal freezing along the margins, decreasing the hydrologic connectivity and enabling cryoconcentration within remnant subglacial liquids to the point of opal precipitation. We illustrate how millennial climate cycles may lead to shifts in subglacial hydrologic connectivity using a Reduced-Complexity Model of Ice Sheet Thermodynamics (RCMIST; Supplementary Note 3). Switches between subglacial melting and freezing are controlled by the basal thermal energy balance, which is comprised of two heat sources: geothermal heat and shear heating, and one sink: conductive heat loss. Therefore, one of these three parameters must change on millennial timescales to elicit the observed hydrologic response. There is no physical reason for geothermal heat flow to vary on millennial timescales, hence we treat it as invariable. Variation in surface temperature accompanying AIM cycles could affect conductive heat loss, but the ~1500 m of ice in the sample source areas[51] would severely dampen these signals and cause a significant time lag for their transfer to the ice sheet bed (Supplementary Note 3). Thus, we infer that shear heating is the most promising mechanism for driving millennial-scale freeze-flush cycles.

Following the simplifying assumption that shear heating can be attributed to ice motion at or near the basal interface[52], we identify two glaciologic variables—ice surface slope and ice thickness—that both drive shear heating and can change on millennial timescales. Changes in ice surface accumulation can drive ice thickening and basal melting

during millennial warm phases, and ice thinning and basal freezing during millennial cold phases. However, ice cores proximal to our sample collection sites show a minimal change in accumulation rate above their noise floor of -0.01 m/yr during AIM cycles[53–56], and detailed records from ice proximal to EM show no change in accumulation during AIM events at the end of the last glacial period[57]. The only millennial-scale variations in accumulation rates clearly resolved in ice core records occur in WDC[58], which is influenced by maritime climate and is not representative of our two sample collection sites located at the edge of the EAIS polar plateau. Therefore, we discount the accumulation-driven model as an unsatisfactory explanation for our observation and favor the ice-dynamical mechanism to explain the observed millennial-scale subglacial hydrologic response (Supplementary Note 3). Ice sheet models[9] show that the tendency for the ice sheet to thicken during warm periods with higher accumulation rates (e.g., interglacials) is overcome by an increase in the dynamic ice thinning associated with grounding line retreat in response to Southern Ocean warming. The dynamic effect driving ice sheet evolution in response to ocean-thermal forcing on grounding lines is incorporated into our simplified model of shear heating through the ice surface slope, which steepens when the ice in the Ross Embayment thins during grounding line retreat (AIM warm phases) and becomes shallower when ice sheet thickness in the Ross Embayment increases during grounding line advances (millennial cold phases). Our model framework assumes that ice thickness at the foothills of the TAM, which is by itself driven by the position of the grounding line in the Ross Embayment, is a linear function of the isotopic records of climate from either WDC (MA113) or EDC (PRR50489) ice cores (Figs. 2c, 3d). Changes in this ice thickness feed into variations in ice surface slope, basal shear stress, ice velocity, and basal shear heating, which affect the basal heat budget in the TAM.

Calculated basal freezing/melting rates in the two inferred regions of sample formation provide a satisfactory visual match to the radiometrically dated records of calcite and opal precipitation from PRR50489 and MA113 (Figs. 2e, 3e). This result is consistent with linear sensitivity of ice thickness in the Ross Embayment to the climatic variations reflected in isotope proxies in AIS ice core records, which are dominated by variability in ocean conditions with additional impacts of atmospheric temperature changes[59]. Scaling laws for ice sheet volumes[60] indicate a high sensitivity of ice volume to ice thickness changes, implying that the volume of AIS exhibited non-linear sensitivity to the millennial-scale climate forcing recorded in Antarctic ice cores. Collectively, our paired ice sheet thermodynamic simulations and precipitate records demonstrates that the ice on the EAIS side of the Ross Sea experiences significant thickness and volume fluctuations not only in response to large climate warming events during glacial terminations, but also in response to climate cycles that are both smaller in amplitude and shorter in duration than major terminations, such as AIM events. Ice thickness changes forced by orbital variations in global temperature[61] may amplify this subglacial hydrologic response beyond what is observed from our subglacial precipitate record, which does not include a glacial termination.

To produce basal freeze-melt cycles that are temporally correlated with precipitate opal-calcite transitions (Figs. 2e, 3e), the RCMIST requires changes to ice thickness of a few hundred meters at the foothills of TAMs (e.g., near the mouth of the valley containing David Glacier for PRR50489) (Supplementary Figs. 9, 10), a small fraction of the -1 km of post-LGM ice drawdown in the TAMS[62,63]. This forcing propagates through outlet glaciers on a timescale of 1 kyr, causing ice thickness changes of dozens of meters in precipitate source area at the edge of the EAIS plateau (Supplementary Note 3). Ice sheet model runs simulating AIS response to millennial-scale ocean thermal forcing support the possibility of thickness change on this scale throughout the Late Pleistocene[9]. Our results imply that ice around the Ross Embayment exhibits a high sensitivity to millennial-scale ocean thermal forcing

during both glacial and interglacial background climate states. On this basis, ice drawdown of a magnitude ca. 10–20% of the total LGM to modern thinning[62,64] is possible during AIM events throughout the Late Pleistocene. While dating uncertainties in AIS precipitates prevent us from assessing leads or lags in ice response to climate forcing, the agreement between opal-calcite transitions and Southern Hemisphere millennial climate cycles implies synchroneity (within dating uncertainties between 1 and 3 kyr) between millennial climate cycles and ice sheet thickness changes.

Results from our RCMIST indicate that opal-calcite transitions are triggered by ice velocity changes generated by ocean-atmosphere teleconnections. Rather than a linear response to smooth climate forcing, the precipitate record describes millennial-scale ice sheet motion governed by thresholds in ocean and atmospheric temperature, corroborating numerical models for ice sheet behavior in AIS embayments[65]. Although many unconstrained physical parameters of the ice sheet system make direct quantification of climate thresholds beyond the scope of this manuscript, we can compare opal-calcite timeseries data with ice core records to place approximate bounds on the climate state required to elicit a millennial-scale ice sheet response. Our RCMIST simulations describe a minimum ice sheet thickness change –which corresponds with a value in ice core isotopic records needed to generate a millennial-scale subglacial hydrologic response (Supplementary Eq. 4; Supplementary Figs. 9, 10). We test the viability of this approximate climate threshold value (Figs. 2c, 3d) by calculating the probability of calcite precipitation above it and opal precipitation below it. For a record completely unrelated to climate threshold forcing there will be an equal probability of calcite or opal above or below the climate threshold, resulting in a 50% success rate. Samples MA113 and PRR50489 return 97% and 85% match respectively, between calcite above the climate threshold and opal below it. This consistent relationship corroborates the idea that climate forcing is responsible for opal-calcite transitions, and points to a dynamic ice sheet response to threshold climate forcing as the triggering mechanism of ice sheet velocity and volume changes.

## Discussion

The results presented here indicate that two rock samples each consisting of opal and calcite layers, separated by -900 km, and deposited tens of thousands of years apart, formed as a result of cyclic subglacial hydrologic processes that match the timing of Southern Ocean and Antarctic climate changes recorded in ice cores, implying a link between AIS basal conditions and Southern Hemisphere climate. More specifically, the sites of precipitate formation oscillated between the freezing of local brines sourced local to the precipitate formation area during cold periods, and the influx of far field EAIS meltwaters during warm periods. Predictions of modern subglacial thermal conditions[66,67] indicate that large portions of the EAIS are at or near the pressure melting point, with widespread melting in the ice sheet interior and freezing along the ice sheet periphery (Fig. 1a). Comparisons between simulations of modern basal melt rate[67] and precipitate collection sites (Fig. 1b, c) indicate that both PRR50489 and MA113 are found within 10 km of the regional boundary between basal freezing and melting predicted by an ice sheet model with 5-km horizontal resolution[67]. Given the evidence suggesting that the depositional area of the precipitates was within 10 km of their collection sites, PRR50489 and MA113 likely formed close to this subglacial freeze-melt boundary. We therefore propose that the cycles of basal melting and freezing indicated by opal-calcite precipitates are the result of migrations of the basal thermal and hydrologic boundary, causing changes in the connectivity between waters from the interior and edge of the ice sheet following millennial-scale climate cycles (Fig. 6). This finding suggests that the subglacial hydrologic response to climate forcing propagates from near ice sheet margins to the ice sheet interior during climate change events.

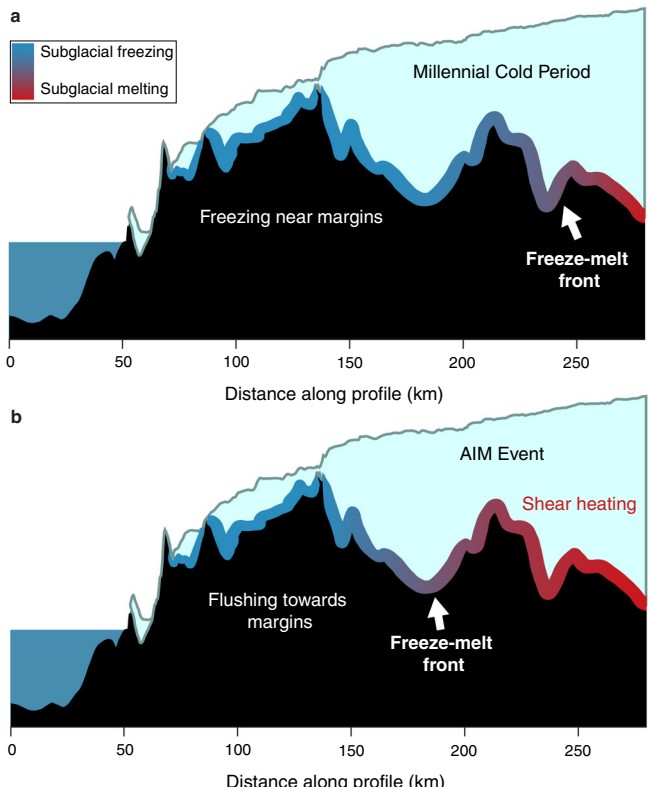

**Fig. 6 | Schematic of subglacial hydrologic change during millennial climate cycles. a** Schematic of subglacial hydrologic systems during a millennial-scale cold period. During this time marginal aqueous systems are frozen and isolated from interior meltwater inputs. **b** Schematic of subglacial hydrologic systems during a millennial-scale warm period (AIM). During these events accelerated ice drives basal shear heating, which allows the basal freeze-melt front to migrate towards the ice sheet margin. This process allows subglacial meltwater from the ice sheet interior to flush towards the margin. Total distance of horizontal migration of the freeze-melt front is unknown but must be >10 km based on precipitate collection location and inferred location of formation.

Based on our RCMIST, the most parsimonious explanation for changes in subglacial hydrologic connectivity is acceleration of ice flow during AIM phases. The driving mechanism for ice sheet acceleration on these timescales is generally regarded to be grounding line migration stemming from ocean-thermal forcing on ice shelves and grounding lines[68]. During millennial cold periods, grounded ice advances towards the continental shelf edge. As the bipolar seesaw takes effect, the ocean-atmosphere teleconnection between slowing AMOC and strengthening Southern Hemisphere westerly winds drives upwelling of relatively warm circumpolar intermediate waters[59], which contact ice shelves and grounding lines. Ice shelf thinning reduces back stress and increases ice discharge across the grounding lines[4], leading to gradual catchment-scale ice-flow acceleration[7]. Corresponding ice thinning starts near grounding lines and propagates upstream, leading to steepening of surface slopes and increased driving and basal shear stress. Higher ice-flow rates and basal stress increase basal shear heating, which triggers enhanced basal melting and subglacial hydrologic connectivity. Proxy records for Southern Ocean sea surface temperature[69] and upwelling[70] provide evidence for millennial-scale variations in-phase with the bipolar seesaw. This finding suggests that Antarctic ice volume is sensitive to ocean forcing on millennial timescales during all background climate states, providing a conceptual framework for assessing future ice mass loss and for interpreting distal evidence for sea-level high stands during Quaternary warm climate periods[71].

The basal thermal regime of AIS outlet glaciers is highly complex, with models demonstrating along-flow transitions between frozen and unfrozen basal conditions resulting from variations in bed topography, ice thickness, and flow rate along the ice flowlines[72]. While we acknowledge that localized basal temperature change could affect precipitate mineralogy, the collective geochronological and geochemical dataset presented here strongly favors hydrologic cycles driven by regional, rather than local, ice response. The consistent relationship between subglacial transitions from freezing to melting recorded at two distant locations over a combined timeframe of over 100 kyr requires a highly regular triggering mechanism that is linked to the broader climate system. On both a glacier and regional scale, temporal fluctuations between basal freezing and melting in the regions immediately upstream of these sample locations necessitate a change in ice sheet dynamics, as atmospheric temperature change could not propagate to the ice sheet base on millennial timescales (Supplementary Note 3) and millennial-scale variations in surface accumulation are either nil[53–56] or temporally inconsistent[58]. This requisite dynamic ice sheet response implicates grounding line migration and ocean forcing as the mechanism driving the observed millennial-scale subglacial hydrologic changes regardless of spatial extent. Based on the locations of the two chemical precipitates studied here, an ocean-cryosphere teleconnection must operate in two ice catchments that are separated by ~900 km and are not part of the same ice drainage basin, pointing to an embayment-wide ice mass and grounding line fluctuation on millennial timescales. Geochemical evidence for millennial-scale flushing of dome-like meltwaters to marginal locations (Fig. 4) suggests that ice sheet acceleration in response to ice shelf perturbation enhances hydrologic connectivity between subglacial waters separated by hundreds of kilometers. Given these spatiotemporal constraints, we conclude that opal-calcite transitions in subglacial precipitates result from millennial-scale migration of the regional freeze-melt boundary beneath grounded ice around the Ross Embayment.

The strength of AMOC is modulated by Northern Hemisphere ice volume and global $CO_2$, such that millennial climate cycles achieve maximum magnitude and frequency during intermediate climate conditions, and are weak during peak glacial or interglacial periods[20]. Current records of AIS mass loss on millennial timescales[21–23] are interpreted to result from Southern Ocean forcing. Yet, these records are confined to glacial terminations leaving it unclear whether ocean temperature drives AIS mass loss outside of orbitally driven warm periods, and whether lower magnitude interglacial ocean forcing can elicit an AIS response. The subglacial precipitate record and model outputs presented here suggest that millennial-scale subglacial hydrologic changes, and ice mass variation are triggered by grounding line migration and ice thickness changes at the mouth of TAM outlet glaciers in the Ross Embayment. This dynamic ice response requires a high sensitivity of the AIS to ocean-thermal forcing regardless of the background climate state, suggesting that ice at the Antarctic margins responds dynamically when a threshold in ocean forcing is reached. Archives of Southern Ocean upwelling rate demonstrate changes in upwelling intensity during both millennial and orbital cycles[70], driving contemporaneous ocean temperature cycles[69]. Our results, when combined with models for AIS loss during glacial terminations[6], demonstrate that resulting ocean-thermal forcing drives ice loss that outpaces any increases in accumulation rate, exerting dominant control of over ice dynamics and mass balance on both millennial and orbital timescales.

## Methods

### Subglacial precipitate opal-calcite timeseries

Timeseries describing mineralogic shifts between opal and calcite in two subglacial precipitates are derived from $^{234}U$-$^{230}Th$ ages combined with elemental characterization (Supplementary Methods). Accuracy

of the uranium method is evaluated using Uranium standard NBS4321 (Supplementary Fig. 8). To construct the stratigraphic age model for each sample, we input sample height and $^{234}$U-$^{230}$Th dating dates into a Bayesian Markov chain Monte Carlo model that considers the age of each layer and its stratigraphic position within the sample to refine the uncertainty of each date using a prior distribution based on the principal of superposition[32]. Elemental maps showing calcium and silicon concentration (Figs. 2a, 3a, b) were produced using Energy Dispersive X-ray Spectroscopy (EDS) measured on the Thermoscientific Apreo Scanning Electron Microscope (SEM) housed at UCSC. EDS data were generated using an Oxford Instruments UltimMax detector and were reduced using AZtecLive software. To quantify the opal-calcite transitions in the samples, Si and Ca concentration data were produced from line scans across precipitate layers (Figs. 2a, 3a, b). For sample MA113, detritus within two calcite layers results in Si peaks that do not correspond to opal. These areas are identified by high aluminum concentrations and are corrected to reflect a calcite composition. Timeseries in Figs. 2b, 3c were then generated by plotting the Bayesian stratigraphic age model, against Ca concentration spectra. The linear relationship between the opal-calcite timeseries and ice core climate proxies implies synchroneity between precipitate mineralogic changes and Southern Hemisphere millennial climate. However, dating uncertainties in our U-series ages are on the order of 1 kyr and in the ice core record are -0.5 kyr[73], thus we are not able to quantify sub-millennial leads or lags between the AIS response and climate cycles. Nonetheless, stratigraphic consistency between dated layers, the regular frequency of mean ages, and the significant correlation between our mineralogic timeseries and climate proxy records supports our conclusion of a link between climate teleconnections and subglacial hydrology. Furthermore, calcite layers form rapidly upon introduction of carbon-rich, alkaline waters from the EAIS interior to the marginal system, and the system then slowly transitions back to opal precipitation after hydrologic connectivity is shut off and the waters freeze. Therefore, it is possible that there is missing time between calcite layers that is not accounted for in stratigraphic age models. However, based on the regularity of opal depositional cycles, and the similarity between precipitate opal-calcite cycles and climate proxies, these unconformities do not represent enough time to disrupt the millennial-scale cyclicity of the precipitate mineralogy.

Correlation between opal-calcite timeseries from both samples and ice core climate records is assessed both visually, and by testing the probability that opal-calcite transitions respond to the crossing of a climate threshold. Although the link between subglacial hydrologic events and Southern Hemisphere climate cycles is the result of a complex ocean-atmosphere-cryosphere feedback, stacked records reveal a clear overlap between the mineral transitions in precipitates and ice core climate proxies on a millennial timescale (Figs. 2b, c, 3c, d). We test the relationship between climate forcing and subglacial precipitate mineralogic transitions by setting a climate threshold based on the minimum ice thickness change required to elicit a millennial-scale response in the subglacial environment as described by our RCMIST (Supplementary Eq. 4). We then calculate the probability that calcite is precipitated at temperatures above this threshold and opal is precipitates at temperatures below it. As a first step in this threshold calculation, we use a Monte Carlo simulation to randomly create 10,000 possible sample accumulation histories within the uncertainty bounds defined by our age models. We identify a best fit precipitate timeseries based on which accumulation model results in the best match of calcite above and opal below the defined temperature threshold.

## Stable isotopic analyses

Carbonate isotope ratios ($\delta^{13}C_{CO3}$ and $\delta^{18}O_{CO3}$) were measured by UCSC Stable Isotope Laboratory using a Themo Scientific Kiel IV carbonate device and MAT 253 isotope ratio mass spectrometer.

Referencing $\delta^{13}C_{CO3}$ and $\delta^{18}O_{CO3}$ to Vienna PeeDee Belemnite (VPDB) is calculated by two-point correction to externally calibrated Carrara Marble 'CM12' and carbonatite NBS-18[74]. Externally calibrated coral "Atlantis II"[75] was measured for independent quality control. Typical reproducibility of replicates was significantly better than 0.05‰ for $\delta^{13}C_{CO3}$ and 0.1‰ for $\delta^{18}O_{CO3}$.

To measure organic carbon isotope ratios ($\delta^{13}C_{org}$), inorganic carbon (IC) was extracted with 1 M buffered acetic acid (pH 4.5), followed by repeated water rinses to completely remove the buffered acetic acid and residual cations from the sample IC. These IC-extracted sample residues were then freeze-dried, weighed, encapsulated in tin, and analyzed for carbon (C) stable isotope ratios and concentrations by the University of California Santa Cruz Stable Isotope Laboratory using a CE Instruments NC2500 elemental analyzer coupled to a ThermoScientific DELTAplus XP isotope ratio mass spectrometer via a ThermoScientific Conflo III. Measurements are corrected to VPDB for $\delta^{13}C$. Measurements are corrected for size effects, blank-mixing effects, and drift effects. Typical reproducibility is significantly better than 0.1‰ for $\delta^{13}C_{org}$.

Opal layers were analyzed at the Stanford University Stable Isotope Biogeochemistry Laboratory for $\delta^{18}O_{SiO2}$ by conventional BrF$_5$ fluorination (e.g., refs. 76,77) and measured with O$_2$ gas as the analyte on a ThermoScientific MAT 253+ dual-inlet isotope ratio mass spectrometer (IRMS)[78,79]. Briefly, 2–3 mg opal samples were loaded into nickel reaction tubes and heated for 2 h at 250 °C at high vacuum. Samples were then repeatedly pre-fluorinated at room temperature with 30 mbar aliquots of BrF$_5$ until <1 mbar of non-condensable gas was present. A 30× stoichiometric excess of BrF$_5$ was added to the nickel tubes and sealed. The nickel tubes were then heated at 600 °C for 16 h to quantitatively produce O$_2$. The generated O$_2$ gas is then sequentially released into the cleanup line, cryogenically cleaned and frozen onto a 5 Å mole sieve trap immersed in liquid nitrogen, equilibrated at room temperature with the IRMS dual-inlet sample-side bellows and measured for $\delta^{18}O$ against a reference tank of known $\delta^{18}O$ composition (24.3‰). Opal $\delta^{18}O$ is reported based on daily corrections made to four primary silicate standards (NBS-28, UWG-2, SCO, and L1/UNM_Q, which are quartz, garnet, olivine and quartz, respectively), spanning ~13‰, and have been recently calibrated to the VSMOW2-SLAP2 scale[77,80]. Three secondary standards (BX-88 (Stanford Laboratory internal standard), UCD-DFS (obtained from H. Spero, UC Davis; values reported in ref. 81) and PS1772-8 (obtained from J. Dodd, Northern Illinois University; measured at U. of New Mexico and reported in ref. 82, which are quartz, opal-CT and opal-A, respectively) were also analyzed over the course of the analyses. Replicate measurements of standards demonstrate reproducibility of <0.3‰ for all secondary and primary standards except the PS1772-8 standard, though heterogeneity in this standard is suspected with laboratory averages reported in the literature[82] ranging from 40.2 to 43.6‰ (average value of 41.5‰ in this study).

## Sr isotopic analyses

Sr isotopic measurements were made at the UCSC Keck Isotope Laboratory. Sr compositions are measured on a TIMS in a one sequence static measurement: $^{88}$Sr is measured on the Axial Faraday cup, while $^{87}$Sr,$^{86}$Sr,$^{85}$Rb, and $^{84}$Sr are measured on the low cups. Accuracy of the $^{87}$Sr/$^{86}$Sr measurements is evaluated using Sr standard SRM987 compared to a long-term laboratory average value of 0.71024, with a typical reproducibility of ±0.00004.

## LA ICP-MS methods

Laser ablation inductively coupled plasma–mass spectrometry (LA ICP-MS) analyses were conducted at the Facility for Isotope Research and Student Training (FIRST) at Stony Brook University. Analyses were made using a 213 UV New Wave laser system coupled to an Agilent 7500cx quadrupole ICP-MS. The National Institute of Standards and

Technology (NIST) 612 standard was used for approximate element concentrations using signal intensity ratios. Laser data were reduced in iolite[83]; element concentrations were processed with the trace-element data reduction schemes (DRS) in semiquantitative mode, which subtracts baselines and corrects for drift in signal.

## Geochemical models of mineralogic cyclicity in subglacial precipitates

To understand the conditions under which discrete pulses of opal and calcite are precipitated following cold and warm Antarctic climate periods respectively, we integrate geochemical and isotopic characterization of the precipitates to inform simulations run using the aqueous geochemical modeling program PHREEQC[49]. The high ion concentrations of subglacial fluids necessitates the use of the Pitzer specific ion interaction approach, which allows PHREEQC to model the aqueous speciation and the mineral saturation index of brines, and has been shown to yield results[44] consistent with the subzero database FREZCHEM[84]. Our modeling approach to simulating opal-calcite transitions can be describe in three parts: (1) Identify the water composition and conditions under which opal will precipitate and calcite will not; (2) Identify the composition and volume of water required to mix with opal-forming fluids to produce calcite; (3) Utilize the $\delta^{18}O$ and $\delta^{13}C$ isotopic composition of calcite and opal, along with the known or inferred composition of mixing waters (Fig. 4) to constrain the relative volumes of brine and meltwater, thereby testing the validity of the mixing model. While the exact ionic strength of subglacial fluids and temperature of the subglacial aqueous system is unknown, we outline a plausible scenario for discrete layers of opal and calcite that fit modeled conditions at the base of the ice sheet[66], and the geochemical constraints measured in precipitates.

Opal precipitation occurs when a solution is saturated with respect to amorphous silica. Opal solubility is both temperature and pH dependent[85], with lower pH favoring precipitation. The silicon concentrations of subglacial waters[86] and mature brines that emanate from ice sheets[87] are typically tens of ppm,−values similar to other surface waters[88]−and Si concentration does not scale with total dissolved solids[89]. At these relatively low Si concentrations, saturation of amorphous silica cannot be achieved without a mechanism to concentrate Si in solution. For aqueous systems beneath an ice sheet, this mechanism is very likely cryoconcentration via subglacial freezing, which extracts water from the cavity at the base of the ice sheet, concentrating solutes and raising mineral saturation[29,90,91]. Yet, most surface waters upon reaching saturation of amorphous silica will also be at saturation for calcite[29,91], thus not matching the discrete opal layers observed in our precipitates. This suggests that the opals form from a mature brine[89] that is relatively free of $HCO_3^-$. We describe mixing relationships between Sr, C, O, and U isotopic compositions, along with Ce* values, that match subglacial fluids observed beneath the EAIS. A candidate fluid that fits our compositional criteria are $CaCl_2$ brines, which occur in the McMurdo Dry Valleys (MDV) as shallow subsurface waters[92–95], deep groundwaters[44,94,96], and surface waters[97–99], most notably feeding Lake Vanda[87,100] and Don Juan Pond(DJP)[96,101,102]. Ca-Cl-rich brines also occupy regions that were previously covered by the North American[89] and Fennoscandian ice sheet[103], implying that they are a natural product of fluid isolation beneath ice sheets. Therefore, $CaCl_2$ brines are a plausible composition for brine beneath the EAIS. The most geochemically well characterized MDV brines are those that feed DJP, therefore, we explore opal precipitation by equilibrating DJP brine with ice, calcite, and opal at a range of temperatures between −5 and 5 °C. At full concentration, DJP brine causes melting of the overlying ice due to its exceptionally high ionic strength, resulting in significant dilution of the original solution, which inhibits opal precipitation (Supplementary Fig. 6a). Subsequent simulations of 10x and 50x diluted DJP brine over the same range of temperatures result in a gradual increase of opal saturation due to the

incremental removal of water via cryoconcentration (Supplementary Fig. 6b, c). In these modeled iterations, opal precipitation is controlled by the amount of water removed from the solution by freezing, which itself is controlled by ionic strength. At 10× concentrated, the solution does not freeze enough to drive opal precipitation, while opal is precipitated from the 50× diluted solution between −3 and −4 °C. A fourth PHREEQC model was run equilibrating DJP brine with ice, calcite, and opal over this temperature range, indicating that opal precipitation is reached once ~75% of the water in the original solution is lost via freezing, which occurs at −3.5 °C (Supplementary Fig. 6d).

Simulation of calcite precipitation during AIM warm periods assumes that 'meltwaters' are mixed with the concentrated 'glacial brines' from part 1. The decision to model calcite precipitation using the admixture of new waters, rather than to reverse subglacial freezing, is based on the disparity in geochemistry between calcite and opal-forming waters (Figs. 4, 5). Based on the hypothesis that calcite layers form when waters from the EAIS interior are flushed to the ice sheet edge, meltwaters driving calcite precipitation are likely to have become higher in alkalinity and dissolved ions through water-rock interaction and chemical weathering of silicate minerals in the substrate during long-term storage beneath the EAIS. Calcite and opal data in $^{87}Sr/^{86}Sr$ vs. $\delta^{18}O$ space provide further evidence that the two end-member waters−brine and subglacial meltwater−dissolve silicates with different provenance. Figure 4c, d show $^{87}Sr/^{86}Sr$ vs. $\delta^{18}O$ mixing curve between opals and calcite, which indicate that the brine provides 98% of Sr to the system from a source with an $^{87}Sr/^{86}Sr$ of 0.7135, while the meltwater endmember has a far lower concentration of Sr (2%) derived from a reservoir with an $^{87}Sr/^{86}Sr$ of 0.71. These data are consistent with a brine that weathers silicate minerals over long periods, and a meltwater that drives silicate weathering with a very different provenance over a relatively short time duration. The $\delta^{13}C$ of the calcite (−23‰) (Figs. 4a, b, 5d) also provides evidence for the chemical composition of meltwaters, implying that they accumulated carbon through microbial respiration, which does not fractionate during the transition from aqueous $CO_2$ to $HCO_3^-$ if 100% of the carbon undergoes this conversion. Given, the similarity between $\delta^{13}C$ composition of calcite and that of respired carbon, it is unlikely that dissolution of sedimentary carbonates took place in the meltwater, which would have added a significantly less $^{13}C$-depleted source of carbon to the reservoir. This framework suggests that candidate compositions for calcite endmember fluids are waters with a similar history of subglacial exposure to glaciated sediments. The EAIS waters that best fit this description are jökulhlaup waters[47] observed near Casey Station, Antarctica. Using PHREEQC, we explore mixing of brine from part 1 with jökulhlaup waters from Casey Station[47], by simulating a range of possible mixing ratios between brine and meltwater. We infer, based on the Ca:Sr ratios from other surface water systems that contain $CaCl_2$ brine, that the ratio of Ca concentration between the brine and meltwater matches the ratio of Sr concentration between the two fluids (brine:meltwater = 98:2). Results show that without at least 30% meltwater the mixture too diluted with respect to Ca and $HCO_3^-$ to precipitate calcite. PHREEQC mixing models successfully produced discrete pulses of calcite with mixtures between 30 and 80% meltwater; conditions under which the admixture is undersaturated with respect to opal because the solution is too dilute with respect to Si and is supersaturated with respect to calcite leading to precipitation (Supplementary Fig. 7). Therefore, the addition of carbon-rich, alkaline meltwaters to opal precipitating, Ca-Cl-rich brines can trigger calcite supersaturation driving rapid calcite growth, consistent with our geochronologic outputs and calcite morphology. Collectively this modeling effort, along with the timescale data presented in Figs. 2, 3, and the fibrous crystal textures, suggests that calcite forms rapidly after meltwaters are added to the subglacial aqueous system. However, the relative volume of meltwater added is unclear from these results alone and requires further isotopic constraints outlined below.

For both the calcite and opal oxygen isotope data, the formation water $\delta^{18}O$ (Figs. 4, 5) is calculated using the appropriate equilibrium water-mineral fractionation factors assuming a temperature (T) of 0 °C (273.15 K). For calcite we use the empirical 1000lnα versus 1/T relationship of[104] and for opal the 1000lnα versus 1/T$^2$ relationship of[105] for T in kelvin. The 1000lnα values for calcite and opal are 33.6‰ and 44.2‰, respectively, and we calculate the formation waters avoiding the non-linearity associated with delta notation far from the standard of choice (in this case VSMOW). As such, we calculate the formation waters composition as: $\alpha = (1000 + \delta^{18}O_{mineral})/(1000 + \delta^{18}O_{water})$.

Calcite data, in $\delta^{13}C$ vs. $\delta^{18}O$ space, define a trend that suggests they form through admixture with an $^{13}C$- and $^{18}O$-depleted water. The carbon concentration dependent mixing curve that best fits that calcite data alone requires that the isotopically depleted endmember, what we'll refer to here as the meltwater endmember carries a higher DIC than the isotopically heavier water, which we'll now refer to as the brine endmember. In Fig. 4, we assume that isotopically heaviest opals record the $\delta^{18}O$ composition of the brine endmember and that the $\delta^{13}C$ composition matches marine carbon derived from the substrate (−0‰), the latter of which is recorded by sodic carbonates suspected of forming from brines in the Lewis Cliff area[106]. Under such assumptions the carbon ratio between meltwater and brine is 97:3 for PRR50489 (Fig. 4a) and 80:20 in MA113 (Fig. 4b), a result that is consistent with the calcite precipitation model presented above, whereby the addition of a carbon rich, oxidized meltwater, to a reduced or intermediate $CaCl_2$ brine, triggers calcite precipitation. As shown in Supplementary Fig. 6, the calcite data imply formation when there is >30% of meltwater in the mixture. The array of calcite data can also be fit by a mixing model that assumes an isotopically more $^{13}C$-depleted carbon composition ($\delta^{13}C = -15$ ‰). While feasible, this is a less appealing solution, as a $\delta^{13}C$ of −15‰ does not match the composition of any specific carbon source and would require a mixture of waters. The data presented here suggests that over the -100 ka of aggregate sample precipitation, there are two consistent endmember waters: an intermediate to reduced brine that is locally derived (star in second quadrant, Fig. 4), and an oxidized meltwater that is from the polar plateau (star in third quadrant, Fig. 4).

## Reporting summary

Further information on research design is available in the Nature Research Reporting Summary linked to this article.

## Data availability

The U-series and isotopic data generated in this study have been deposited in the US Antarctic program database and can be accessed at https://www.usap-dc.org/view/dataset/601594. Sample PRR50489 was provided by the US Polar Rock Repository at the. Byrd Polar and Climate Research Center at Ohio State University, and can be accessed at https://prr.osu.edu/collection/object/100478. Source data are provided with this paper.

## Code availability

Codes used in this study are available at https://github.com/GavinPIccione/AntarcticMillennialCycles, and can be accessed and cited at https://doi.org/10.5281/zenodo.6968568.

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

## Acknowledgements

We thank G. Faure for his sample collection at the Elephant Moraine and A. Grunow and the Byrd Polar Rock Repository for providing sample PRR50489. We also thank Warren Sharp for a helpful pre-submission review of a previous version of this work, and G. H. Edwards and S. Hemming for insightful input. This research was funded by the NSF graduate research fellowship to G.P.; NSF 2042495 to T.B., S.T., and M.H.; NSF 2045611 to E.T.R. and P.N.; NSF 0944578 to K.L.

## Author contributions

G.P., T.B., and S.T. conceived of the study. G.P. wrote the paper with contribution from T.B. and S.T. G.P. collected U-series and Sr data. S.T. wrote the RCMIST and made glaciologic interpretations. E.T.R. collected laser ablation data and aided in geochemical interpretations. M.H. aided in PHREEQC modeling and geochemical interpretations. D.E.I. and K.M. collected $\delta^{18}O$ data on opals. G.P. and C.T. performed clean laboratory work. B.C. collected SEM images. P.N. collected XRF data and aided in geochemical interpretations. K.L. collected sample MA113 and aided in geologic interpretation. All authors provided edits to the main text of the paper.

## Competing interests

The authors declare no competing interests.
