## [Peer Review File · Nature Communications]

Subglacial Precipitates Record Antarctic Ice Sheet Response to Late Pleistocene Millennial Climate CyclesREVIEWER COMMENTS

Reviewer #1 (Remarks to the Author):

This research article submitted to Nature Communication is extremely interesting to a broad audience and uses a novel, poorly exploited geological archive - subglacial deposits - to draw links with Antarctic Ice sheet dynamics and millennial climate cycles. This has profound implications for future predictions of the effects of warming climates on ice loss in the white continent and ice sheets destabilization.

Application of geochemistry and geochemical models demonstrate that EAIS subglacial carbonates and silica are important and accurate archives of past environmental and climate changes. This research supports positioning subglacial precipitates at the same level as other continental sedimentary archives and shows their tremendous potential.

There is a wealth of information that was extracted from two specimens of subglacial carbonates, which were spaced, when collected, over 900 km apart. These two samples cover a time span, as determined by radiometric dating, of ca 230 to 140 Ka (PRR50489) and 54 to 42 Ka (MA113), which allows detecting millennial changes in the subglacial environment over broadly two glacial cycles.

Research papers such as this one, in the opinion of this reviewer, merit to be published in a high impact journal such as Nature Communications, pending some improvements. Specifically, the dataset and results need more work. I, therefore, just looked at missing data and missing insight on mineralization processes, as the discussion and conclusion should come after the points I am proposing are addressed. I am aware that these points may be outside the field of geochemistry and palaeoclimate, but palaeoclimate proxies extracted from geological archives need an understanding about how the geological archives capture the proxies. And also how the geological archive proxy capture may be influenced by the geology.

Major points

1) Geologic/Geomorphologic setting

The samples, to my understanding, were collected in moraines, not in-situ. In relation to the travel distance of the samples, I agree it is not hundreds or thousands of Km, but there is no certainty it is <50 km, even by looking at Fig. 2 (in panel b and c the locations of the samples overlap with the pixels of the melt rate). In line 259 it is stated that the modern ice flow for the Elephant Moraine (EM), is low. I checked reference 55: that paper provides a range of 5 to 2.5 m/year, but it is the region labelled NW1 that has low ice flow. My attention was caught up by this sentence: "meteorites falling onto the "fast flow" area will ...strand at the Elephant Moraine".

Sample MA133 was collected in the Mount Archenar moraine, where some exposure ages point to a broad ice-sheet thickening and provenance change of the material constituting the moraine from ca. 50 to 12 Ka (Bader et al., 2017).

The geomorphological contextualization of the moraines is worth being, at least, briefly mentioned in the framework of the discussion. The inferred geologic framework is existing for both the Elephant and the Mount Archenar moraine sites (Faure et al., 1998; Bader et al., 2017). This would provide important information because the mineralogy of subglacial precipitates depends on their bedrock source. See my comments in the following points where other subglacial deposits in Antarctica are mentioned.

Finally, a recent paper on the Mount Archenar moraine states that “ the similarity to Lake Untersee suggests commonality between Mt. Achernar Moraine with other Antarctic sites where brine formation is minimal.” (Graly et al., 2020). Could this be discussed?

2) Petrographic considerations

Petrography is entirely missing, apart from images of polished slabs. Here are some points relevant to the results and discussion:

a) From Extended Data Fig. 1, the PRR50489 and MA133 it appears that they have different petrographic characteristics. MA133 looks more similar to a true subglacial precipitate and PRR50489 looks more like what Faure et al's (1988) inferred it to be: a subglacial travertine dislocated by subglacial tectonics. Also, MA133 appears to have two different facies, one consisting of brown, coarse calcite and opal lenses and one consisting of darker, fine grained (?) calcite and opal. This may signify a major change in the environment or a different provenance of the fluids...also for the calcites.

PRR50489, I believe, should come from the suite of samples collected by Faure and colleagues in their explorations. On the basis of Sr, O and C isotope ratios and trace element concentrations Faure et al. (1988), proposed that the Elephant Moraine subglacial calcite-opal deposit built up in “hot-spots” that gave rise to subglacial springs, where travertine-like deposits formed. The EM sample illustrated in Faure et al (1988) Figure 2 (Sample 582) looks very much like a well-laminated spring deposit. Similarly, PRR 50489 “aspect” is more similar to that of a travertine than “classical” subglacial calcites, with their fluted and relatively delicate morphologies.

Fluted morphologies characterize two other specimens of Antarctic subglacial carbonates (aragonites mostly) illustrated by Aharon (1988), which were sampled on gneissic rocks of the Vestfold Hills and the Boggs Valley calcites, which were also collected in-situ on amphibolites. Both these two examples of Antarctica subglacial carbonates are different petrographically and mineralogically from the PRR 50489 moraine sample. The MA 133 seems to show sparite and microsparite like the Boggs Valley samples in its lighter part, but then it becomes different (Extended Data Fig.1, panel c).

As neither PRR50489 nor MA 133 were “in-situ”, it becomes hard to tell if they were ice-flow oriented subglacial deposits formed from a film of fluid confined by regelation ice (sensu Hallet, 1976; Siman-Tov et al., 2021) or not. Siman-Tov et al., (2021) provide a key for the development of similar subglacial deposit, which I believe is the hypothesis embraced in the present manuscript. This mechanism of subglacial deposits formation implies one type of water flowing through interconnected subglacial conduits (which formed the calcite) and the refreezing of subglacial meltwater occurred and formed opal. The transport of water through conduits transports, however, should carry lithic fragments and fragments of broken subglacial deposits itself, which then would settle from the fluid and become entrapped in the regelation step (Frisia et al., 2017; Siman-Tov et al., 2021). It then stands to reason that at the boundary between the calcite and the opal (transition from open to closed system), the detritus should settle, particularly in furrows. However, this is not commonly observable in the samples. Detritus should also be present at the top of the opal, marking the onset of flow in the open system.

The hypothesis that open system fluids form calcite and refreezing of basal meltwater form opal should petrographically result into: i) formation of microsparite at the lower contact between opal and calcite (cf. Sharp et al., 1990; Frisia et al., 2017); ii) embedding of detritus in the microsparite; iii) re-nucleation of calcite above the opal followed by geometric selection.

These petrographic observations would provide proof of concept.

Microsparite seem to occur in the MA 133 polished slab illustrated in Fig 1 of the Extended Data, but there are two very distinct upper opal lenses that appear to be in direct contact with sparite at the bottom. I use the term “lenses” because the opal layers commonly “taper” out, which is very clear from Fig. 1 in the Extended Data.

The bottom of the opal lenses is almost always concave and the top is flat (even in the PRR sample, where they may be, however, deformed by glaciotectonics). Similar morphologies are hard to find in the literature of subglacial carbonates where I found visual illustrations of the petrography (Bjaerke and Dypvik, 1977; Aharon, 1988; Sharp et al., 1989; Refsnider et al., 2014; Lipar et al., 2021). The only one I could find is Siman-Tov et al. (2021), where thin sections and chemical maps show Si-rich (detritus-rich) layers and calcite layers having flat contacts both at the bottom and the top of the layers. But not a concave bottom and a flat top and detrital-rich layers are common, whilst similar detrital-rich layer are impossible to discern from the slabs in Fig. 1 ED.

Now, the concave lower boundary of opal results from the occurrence of calcite botryoids. The opal may even taper out against calcite. Now, I am wondering if that calcite is heteropic with the opal or not. It would be worth dating.

The visible stratigraphy suggests that opal deposited in furrows between calcite botryoids, rather than in distinct layers. And there is no evidence that the crystal tips of the calcite are corroded. The Si synchrotron map, despite the large pixels, is good enough to suggest that the calcite underlying opal has pristine crystal terminations (PRR50489). Now, would it be a possible alternative explanation that calcite precipitation continued at the tip of botryoids exhausting most of the Ca in solution and furrows hosted micro-reducing systems because of the complete exhaustion of O by subglacial microbial metabolism? I was wondering this also because there are some opal layers with no Ce* anomalies and some calcite

layers with Ce* anomalies (fig. 4, Extended data)? Maybe the system shifted laterally, as it is possible in the complex topographic substrate of the subglacial environment, rather than shifting through time. This does not contradict the hypothesis of externally forced subglacial system changes, it would put it in its environmental perspective, where lateral changes are common (see, for example, the lateral variability of coeval subglacial samples illustrated by Frisia et al., 2017)

The teleconnection hypothesis could better support the existing data particularly given that only one sample has good ages and, overall, the whole interpretation is based on one sample.

b) Another important point for petrography to be carried out is from

Faure et al's (1993). In discussing the high Sr isotope ratios in their non-marine carbonates they suggested possible contamination with fine detritus (clay). I am wondering if the presence of nano- sub-micrometre sized particulate could be an issue also in the two samples analysed here. Could particulate contained in the calcites not be picked up by EDS and have contributed to their Sr isotope ratios? Note that the sample from the Elephant Moraine analysed by Faure et al (1998) has high $^{87}\text{Sr}/^{86}\text{Sr}$ isotope ratios, which are in the range of the samples measured in the present research and in the range of minerals formed from fluids that came into contact with the Beacon Supergroup of the Central Transantarctic Mountains (cf. Faure et al., 1993). This is why I believe that a geologic contextualization is worth looking into.

3) Diagenetic considerations

It is most likely that opal had a precursor. I suppose that opal in the PRR and MA samples is now Opal CT, thus, most likely there was a transformation from Opal A to Opal CT and, perhaps, from a polymerized colloidal suspension to Opal A to Opal CT as it happens in warm springs or in discrete silicification of cave carbonates. The hydration state of the constituents of opal and Ostwald ripening may have a role in kinetic uptake of elements and isotope fractionation. There are kinetic effects associated with dehydration and rehydration during the transformation. REE, for example, are preferentially uptaken in the hydrated phases within a general reducing environment. Kim et al (2012), for example, suggested that negative Ce anomalies might not always be an indicator of paleo-oxic conditions. For REE fractionation, a positive Ce anomaly may appear in particulate organic material and negative Ce anomaly may result from Ce scavenging in marine examples (Haley et al., 2004). Thus, may it be possible that the Ce* is localized and that also some calcite may have positive Ce*? This potential role of diagenesis should be considered in the discussion and either accepted as complexity or discarded on some other basis.

Diagenesis may impact other isotope/element exchanges, particularly when the system becomes open again. Mixing of fluids of diverse ionic strengths, particularly brines, is known to cause dissolution and subsequently precipitation of carbonates.

Sample MA 133 is characterized by large uncertainty in the ages (Fig. 2 in Extended data) that are of the order of thousands of years. A few ages even suggest the possibility of reversals. In Fig. 1 main text, the whole uncertainty of the ages should be reported, not the mean point, to show how a shift of a few thousands of years to the right or left of the mean point would not match the WDC delta18O normalized dataset. The correlation is, in my opinion, a bit stretched even assuming “stratigraphic consistency” (Methods). MA133 dataset potentially point to a diagenetic history (which may be also valid also for PRR50489, but with minor impact).

There is a potential for diagenesis for both Opal (following Ostwald’s step rule) and calcite, due to the mixing of the fluids (brine and subglacial meltwater) with different ionic strengths. It would be rather interesting to see what the tip of the calcite crystals really look like in thin sections, and what the top of the opal layers also looks like, where the diverse fluids mix.

4) Some implications of points 1-3 for Isotope ratios data and trace elements

Faure et al (1998) implied that the underlying bedrock of the EM carbonates is the Beacon Supergroup, because of the depleted $\delta^{13}\text{C}$ compositions (-23‰) of the calcites. These were explained as due to the hydrothermal fluid (in the present submission, hydrothermalism is ascribed to reference 45, which is actually cited erroneously and out of context as I will explain later. The correct reference here is Faure et al., 1998) that came into contact with the coal, in analogy to other measured $\delta^{13}\text{C}$ values of calcites associated with coal. If this is the case, the C of the PRR 50489 would not be sourced by subglacial organic matter in a strict sense (which should be organic compounds/microbes) but from Permo-Triassic coal. It is a tiny difference, but it matters. Microbial oxidation of subglacial organic matter of microbial origin would yield more positive values, around -12‰ (Mook, 2006). I could not find how reference (35) supports that PRR50489 $\delta^{13}\text{C}$ values derived from carbon sourced from subglacial organic matter that is oxidized during microbial respiration.

A $\delta^{13}\text{C}$ value of -23‰ suggests a very negative DIC or that the values are highly kinetically fractionated. This is a possibility for subglacial carbonates (cf Skidmore et al., 2004).

Such low value for the DIC species is actually consistent with Faure et al’s (1998) hypothesis of hydrothermal fluids (even if the Author dismiss their hydrothermal fluids) of subglacial melt origin that percolated through the Beacon Supergroup in depth, became heated and stayed in contact with its lithofacies for a long time. I am not sure that the argument presented in this research paper rules out Faure et al’s 1998 “hydrothermal” fluids. Neither is a good argument if considering Frisia et al. (2017) because their inference was that the calcites mostly precipitated from interconnected subglacial meltwaters. Closure of the system was documented by microsparite, which may be the equivalent of Opal in the EM and AM samples.

There is an alternative: methane oxidation during the precipitation of PRR 50489 calcite.

The MA 113 DIC is 5‰ more positive than PRR 50489, which suggests more variable organic matter oxidation pathways. This should be an aspect to account for in the discussion that may shed light on the provenance of formation waters in the two study sites.

I suppose that Opal is CT. Then, it is most likely that it is a transformation of Opal A. It becomes likely that the $\delta^{18}\text{O}$ signal of opal could reflect kinetics related to such transformation. Structural changes in the silica framework may impart changes in the O isotope composition as observed in marine sediments consisting of opaline silica. During Si-O-Si condensation there may be an enrichment in the heavy isotope. Hence, the suggestion that the mixing model should take kinetics and diagenesis into some “form” of account.

Faure et al. (1988) pointed out that the elevated $^{87}\text{Sr}/^{86}\text{Sr}$ values of the EM calcites are to be ascribed to contact of fluids with “old Rb-bearing rocks of the substrate”. Little is known about the MA samples, unless the bedrock was also Beacon supergroup. Now, the assumption that there is no significant strontium isotopic fractionation for calcite is correct, as it depends on the precipitation rate, which is likely to be slow (cf. Bohm et al., 2012). The spread of the $^{87}\text{Sr}/^{86}\text{Sr}$ values of calcite and opal is, in effect, consistent with various contribution from the Beacon Supergroup. But this assumption is based on the fact that there is no kinetic fractionation associated with the rate of Opal precipitation, the transformation Opal A-Opal CT and the fact itself that opal formation starts with the polymerization of a colloidal suspension. This is a complex nucleation pathway. I was wondering if there are studies that relate Sr uptake in opal and its fractionation in non-crystalline phases of SiO_2 . Most of the knowledge we have is on the crystalline form of minerals. Thus, the potential of transformations should be given insight when explaining the rationale of the model...maybe simply stating that diagenetic transformations were not taken into account but one should be aware that the kinetic processes associated with Ostwald's step rule may influence TE uptake and isotope fractionation.

Line 156-81: Chlorine . The issue here is that chlorine is difficult to accommodate in the lattice of calcite. Cl is a monovalent anion, similar to fluorine. Thus, the two possible substitution mechanisms for Cl in calcite is replacing one oxygen atom within the CO_3^{2-} group to form a $(\text{CO}_2\text{Cl})^-$ group or substitution involving replacement of the CO_3 group by two Cl- ions to form a CaCl_2 defect. The observed uptake of Cl and F decreases from vaterite to aragonite to calcite. Which means that Cl would be accommodated preferentially in a precursor phase in an Ostwald Step Rule series. I am not an expert in the SiO_2 system, but it appears that the hydrated forms of SiO_2 , which may be a precursor phase according to Ostwald Step Rule, uptake Cl. Hence, opal may incorporate more Cl than calcite not because the water has a different provenance, but because opal is hydrated. I suggest this depositional pathway needs to be discussed in the framework of the significance of the PREEQC results. Crystallization pathways and kinetics associated with them are important in palaeo-science.

In this framework, I believe that it is necessary to cite what the mineralogy of the subglacial carbonates found near Carey Station is ...I found it is aragonite. Not just provide a reference, please¹. Interestingly,

Goodwin (1988) reports that “chemical and isotopic analysis of the jokulhlaup water determined its origin as “basal melt water which had been derived from the melting of ice from before the last glacial maximum” and that gained their chemical properties from the interaction with fine grained glacial deposits (which are very reactive). Then, if the mineralogy is aragonite, its formation requires different chemical parameters than those that form calcite. Then, the use of the jokulhlaup events chemistry as an end member for the mixing model may not reflect the actual chemistry of the subglacial meltwaters on the Ross Sea side of the Transantarctic Mountains, which Faure et al. (1998) suggest are influenced by interaction with the Beacon Supergroup. The Geology is rather different and it counts. This is the reason why some subglacial precipitates may show different minerals with diverse trace element concentrations and isotope ratios.

The nature of sediment/bedrock in the Law Dome region, the bedrock in Boggs Valley are not the Beacon Supergroup. Therefore, the idea that formation of subglacial calcite in Antarctica may be related to the same large-scale climate needs, at first, to be tested by contextualizing them in their geologic/geomorphologic setting, with potential fluid composition that evolved from the bedrock. The Law Dome subglacial fluids cannot be taken as end members for the EM and AM precipitates, simply because the minerals that would have been dissolved are very different and because coal in the Beacon Supergroup complicates the geochemical evolution of fluids that go through it (and of the calcites) . Otherwise, the deposits would have similar geochemistry and mineralogy, but they do not.

For example, the evidence from Boggs Valley is that the calcite crusts formed from subglacial meltwater ... not hydrothermal as stated in the present submission... during the Antarctica LGM, as precise ages suggest (see also the latest LGM synchronization between NH and SH by Svensson et al., 2020). The Frisia et al (2017) paper reports that pebbles cemented by calcite may be the result of catastrophic subglacial discharges into the valley from a lake that formed through the influence of volcanism between ca 24 and 22 ky, which coincide with a warm phase recorded by EDML and WDC (Svensson et al., 2020). With the exception of conglomerate-like “layer”, most of their calcite seems to have grown in a cold period. Please amend the citations (Faure et al for the hydrothermal and Frisia et al for the basal ice melting) and put them in the correct perspective.

Minor points:

I would like to point out that the map shown in sample in Fig 1b Extended dataset should have been carried out accounting for the dislocation in the middle of the sample. The direction of growth of the crust changes, so, the top part was analysed along the growth direction, the lower half is at an angle. What is the relationship with the ages of this transect?

The topmost part of MA 133 is different from the part below and appears to have some microsparite associated with calcite and opal. Given the complexity of the stratigraphy, and deformations, it is recommended to plot onto Fig 1 Extended Data where the dating samples were taken. And it would also be commendable to plot the Ca and Si W% for both in terms of distance from top. The plot vs ages in

Fig. 1 of the main text is the interpretation. The raw dataset should be plotted with the microstratigraphy to allow a direct comparison with the petrographic features of the specimens.

Synthesis

I am aware that the only way to solve some issues would be to have reproducible data from at least a second sample of the same age in the EM or MA, so that correlations of stratigraphy and chemistry can be operated. But I am also aware that this may not be possible due to the fragmented nature of moraine specimens.

On the basis of one good sample it is difficult to prove the hypothesis, but I hope I have provided additional tools that may support it as well as some food for thought.

Reviewer #2 (Remarks to the Author):

Piccione et al. investigate a novel paleoclimate archive, a chemical precipitate, over two rather short periods (<100 kyr) during MIS 3 and 7/6, respectively. They argue that the calcium-rich endmember with low d18O values indicates a meltwater source in the interior of the EAIS, while the opal-rich endmember with high d18O values point to meltwaters from a rather marginal ice position. The authors conclude that calcite precipitated in subglacial meltwater during AIM warm periods. They further infer a distinct alteration of opal- and calcite-rich precipitates related to millennial-scale changes, with calcite precipitating during the warmer AIM periods.

I find this an interesting and thoroughly carried out study. The main conclusion, i.e., AIM warm phases show higher delivery of meltwater from the interior of the ice sheet, is intriguing because it contradicts the concept that during a retreat ice melt is primarily enhanced at the edges of an ice sheet. I also find it intriguing that the chemical precipitate exhibits a switch between opal-rich and calcite-rich layers pointing to a threshold behavior in the response, typical for ice sheets, rather than a linear response to a smooth forcing. A third important implication is that the millennial-scale drivers seem to be sufficient to switch the meltwater system, i.e. there is no underlying orbital-scale change required. Even a glacial termination – which is not preserved in the record – could not do more than switching the chemical precipitate. In light of this, I believe this study could be of wide interest.

As far as I can see the investigation is based on sound methods and results, although I need to emphasize that I am no expert in geochemical or ice-sheet modeling. But the chemical analysis, age

dating and age model development seem sound from my point of view. Also, the authors have a clear understanding of how to put their results into a paleoclimatic context. Therefore, it is hard for me to point out any major weaknesses or flaws.

However, I am not expert enough to judge if their main interpretation of precipitate formation is well enough supported, i.e., oscillations between freezing of local brines from proximal sources during cold periods, and influx of far field EAIS meltwaters during warm periods.

According to the results of their ice-sheet modeling, millennial-scale changes are the main drivers of ice elevation changes of several hundreds of meters. This is interesting and in line with independent findings we made based on isotope studies in the Weddell Sea Embayment. It would also imply, as stated by the authors, that the Ross Sea behaved in a similar manner, a fact that has previously been questioned given, for instance, the later response of the WAIS to millennial-scale changes of the last deglaciation.

Even though the time scale is short and the records display primarily millennial-scale changes, I still recommend to display at least an insolation curve (I recommend Southern Hemisphere summer insolation for Dec-Feb at $\sim 75^{\circ}\text{S}$) for PRR50489 in Fig. 1, and discuss the potential impact of orbital forcing on the record. I do see a relation to the model, for instance, in Fig. S1, to both insolation and obliquity. Therefore, simply to avoid being forced to change the interpretation in case a longer precipitate record will be retrieved showing distinct orbital responses, I recommend including the discussion on orbital (ice-age) influence.

On that notion, it would be beneficial if all displays against time contain the MIS stages (as in Fig. 1).

Comments on figures and manuscript arrangement

- I would prefer to see the map as Fig. 1 and data figures after that. It just makes the content more accessible.

- The paper has clearly been written in the short format for, e.g., Nature. However, Nature Com allows for much more content in the main paper. Therefore, I recommend a major re-structuring and re-organization of the paper. Right now there are very extensive supplements for both text/methods and figures, which is confusing and this cluttering is not helping to make a convincing case. The specific changes are up to the authors, however. Subsequently, the supplement should be reduced to one file containing necessary methods and figures that do not fit in the main paper.

As for the additional figures in the main paper I recommend:

- Extended Data Fig. 9 could become the final figure of the main paper.
- Since this is such an unusual climate archive, it would also be beneficial to have Extended Data Fig. 1 – showing the actual precipitates – in the main manuscript.
- Extended Data Fig. 2 contains the age model with the Bayesian model, which would also be appropriate for the main manuscript.
- Extended Data Figs. 3 and 4 can also be displayed in the main manuscript, although they would need to be polished to match the style of Fig. 3.

As for the supplementary methods I recommend to move to the main paper:

- The chapter ‚Subglacial precipitate source areas‘. This is clearly a discussion that fits the main paper.
- The chapter ‚Reduced complexity ice sheet model‘ is right now a mixture of methods and interpretation. This chapter would best be split into the method itself, which should stay in the supplement, and the interpretation, which should move to the main paper. I am not sure about the two associated figures but perhaps Fig. S1 could go to the main text as well (with some orbital curves –see above, and discussion thereof).

More specific comments by line:

- Lines 234-237

If you intend to elaborate on potential leads or lags, you need to discuss your dating uncertainties (the U/Th dates) with the uncertainties of other, Southern Ocean climate indicators. This will be tricky and as you state, beyond your abilities. I would simply mention that the linear relationship implies synchronicity (within dating uncertainties) although you can't quantify leads or lags.

- lines 280-282

What are the implications? Be specific.

- line 349

exception rather than acceptance?

I would be happy to conduct a second review on a structurally revised manuscript.

Michael Weber

Reviewer #3 (Remarks to the Author):

Piccione and colleagues present two subglacial precipitate samples from the EAIS, close to the Transantarctic Mountains. These samples show variations of Ca concentrations at millennial scale variability for two time periods; from 230-150 kyr BP (PRR50489) and 54-42 kyr BP (MA113). These variations coincide with Antarctic ice cores δD signal, where low Ca concentrations correspond to antarctic cold periods and high Ca concentrations to warm Antarctic Isotope Maximum (AIM) peaks. Rich Ca concentrations occur as a consequence of enhanced basal melt production at the interior of the East Antarctic Ice Sheet (EAIS) during warm periods. The novelty of their study is that they claim that these variations in the hydrological cycle are explained through ocean warming, which produces grounding-line migrations. These migrations produce changes in the surface slope affecting ultimately the subglacial heatbudget at the EAIS. If the measured signal can be directly related to changes in the oceanic temperature, then this would confirm the existence of the bipolar-seesaw mechanism caused by changes in the AMOC. Such a finding is well suited for the scope of Nature Communications.

Overall, the manuscript is well written and easy to read for someone as me who is not familiar with subglacial precipitates. Since I cannot give an opinion on the geochemical analysis and the employed dating methods I have only one general comment regarding the mechanism that drives this signal and two technical questions.

General comment:

Atmospheric contribution

The key message of this manuscript relies on the basis that the origin of the signal comes from oceanic variations, however, couldn't this millennial variation be explained through atmospheric warming? It is stated (L292) that such an atmospheric warming could not penetrate to the base of the ice sheet due to insulation effects, however, results from the Greenland Ice Sheet show that atmospheric warming

changes ice dynamics, which increases basal friction and consequently basal melt production (Karlsson et al., 2021). A warmer atmosphere reduces the viscosity of ice sheets and enhances the ice flow, increasing frictional heat production. How can you ensure that the recorded Ca variations occur due to oceanic warming provoking grounding-line migrations and not a warmer atmosphere? Maybe you could do a sensitivity experiment with your ice-sheet model changing the value of “A” (Ice viscosity parameter; Eq. S2) following an Arrhenius law dependent on the temperature.

Technical comments:

Reduced complexity ice sheet model

Which ice sheet model are you using? Is it published? Maybe I understood it wrong, but you don't incorporate any ocean forcing, but rather grounding line migrations to mimic the Ca cycles. Do you have any idea of which oceanic warming you would need in order to produce such a grounding-line migration?

Ice core imprints

Your samples show periodic episodes of basal melt, is this also observable on ice cores close to the grounding line or ice cores close to the sampled regions?

References:

Karlsson, N.B., Solgaard, A.M., Mankoff, K.D., Gillet-Chaulet, F., MacGregor, J.A., Box, J.E., Citterio, M., Colgan, W.T., Larsen, S.H., Kjeldsen, K.K. and Korsgaard, N.J., 2021. A first constraint on basal melt-water production of the Greenland ice sheet. *Nature Communications*, 12(1), pp.1-10.

REVIEWER COMMENTS

Reviewer #1 (Remarks to the Author):

This research article submitted to Nature Communication is extremely interesting to a broad audience and uses a novel, poorly exploited geological archive - subglacial deposits - to draw links with Antarctic Ice sheet dynamics and millennial climate cycles. This has profound implications for future predictions of the effects of warming climates on ice loss in the white continent and ice sheets destabilization.

Application of geochemistry and geochemical models demonstrate that EAIS subglacial carbonates and silica are important and accurate archives of past environmental and climate changes. This research supports positioning subglacial precipitates at the same level as other continental sedimentary archives and shows their tremendous potential.

There is a wealth of information that was extracted from two specimens of subglacial carbonates, which were spaced, when collected, over 900 km apart. These two samples cover a time span, as determined by radiometric dating, of ca 230 to 140 Ka (PRR50489) and 54 to 42 Ka (MA113), which allows detecting millennial changes in the subglacial environment over broadly two glacial cycles.

Research papers such as this one, in the opinion of this reviewer, merit to be published in a high impact journal such as Nature Communications, pending some improvements. Specifically, the dataset and results need more work. I, therefore, just looked at missing data and missing insight on mineralization processes, as the discussion and conclusion should come after the points I am proposing are addressed. I am aware that these points may be outside the field of geochemistry and palaeoclimate, but palaeoclimate proxies extracted from geological archives need an understanding about how the geological archives capture the proxies. And also how the geological archive proxy capture may be influenced by the geology.

We thank reviewer 1 for a thorough review, and for giving a great deal of thought to the underlying mineralogic and geologic processes associated within our samples. In many of these comments, the questions/concerns raised by this reviewer point to areas of the original manuscript that required further explanation of our hypothesis and/or addition of data that we had originally omitted. This reviewer clearly has thought a great deal about subglacial settings, and we appreciate their offering of this expertise to the benefit of our manuscript. We have responded to all points below:

Major points

1) Geologic/Geomorphologic setting

The samples, to my understanding, were collected in moraines, not in-situ. In relation to the travel distance of the samples, I agree it is not hundreds or thousands of Km, but there is no certainty it is <50 km, even by looking at Fig. 2 (in panel b and c the locations of the samples overlap with the pixels of the melt rate).

We do not mean to convey absolute certainty in the sample source, and instead aim to present lines of evidence that support precipitate formation close to the moraine. In the

original manuscript, we included a more detailed description of why we infer a sample source area close to the location of the moraines in the supplementary text. We have added reference to this supplementary text, as well as more supporting text, to the main manuscript (lines 93-105).

In line 259 it is stated that the modern ice flow for the Elephant Moraine (EM), is low. I checked reference 55: that paper provides a range of 5 to 2.5 m/year, but it is the region labelled NWI that has low ice flow. My attention was caught up by this sentence: “meteorites falling onto the “fast flow” area will ...strand at the Elephant Moraine”.

In the supplementary text, we account for this faster flow upstream, which would result in the sample traveling up to 150km. However, we present a most likely scenario, based on the likely slower ice velocities in colds periods during the possible transport time (<145ka based on our youngest age for PRR50489), and the idea the debris traveling in basal ice travels slower than surface ice velocities.

Sample MA133 was collected in the Mount Archenar moraine, where some exposure ages point to a broad ice-sheet thickening and provenance change of the material constituting the moraine from ca. 50 to 12 Ka (Bader et al., 2017).

Our results are broadly consistent with those of Bader et al., 2017, who theorize – based on a lateral moraine intersecting the broad MAM, surface exposure ages between 50-12ka, and morphology of the arch – that the period between 50-12ka represents a period of ice sheet thickening and provenance change.

First, the observed provenance change occurs at, or just before, 55kyr: the oldest age we measure in MA113. The surface exposure ages of 50-12ka measured in the zone of MAM with a broad arch, point to the idea that this arch formed during this time. The morphology of the arch suggests formation “may either be a result of compression from more ice flowing into the moraine during the LGM, or ice sheet thickening where the arch is a remnant of thicker ice from the LGM”. The overall elevation of the arch decreases in the beginning part of the 50-12ka timeframe, followed by an increase in elevation that creates the arch feature. Our ages in MA113 suggest that continuous layers of calcite and opal formed between 55 and 42ka; these layers are interrupted by an unconformity, above which we disregard in terms of our climate record. We dated one calcite layer above this unconformity to have formed at 25ka. It stands to reason that the period of calcite and opal precipitation occurred before major geomorphological restructuring of the moraine. Then during or just prior to the LGM, Law glacier thickened causing the arch to form and disrupting to the subglacial cavity, which led to the observed unconformity.

Finally, the geometry of the MAM, paired with the exposure between 50-12ka, suggest that the site was overall thickened during this time, which is at odds with our idea of ice sheet thickening and thinning causing the opal-calcite transitions. However, our reduced complexity ice sheet model requires ice thickness changes to occur in the mouth of the TAM, which cause ice surface slope changes that drive an increase in velocity. These

thickness changes, however, are not necessarily large around MAM and likely do not disturb the orbitally driven elevation trends in the moraine.

The geomorphological contextualization of the moraines is worth being, at least, briefly mentioned in the framework of the discussion. The inferred geologic framework is existing for both the Elephant and the Mount Archenar moraine sites (Faure et al., 1998; Bader et al., 2017). This would provide important information because the mineralogy of subglacial precipitates depends on their bedrock source. See my comments in the following points where other subglacial deposits in Antarctica are mentioned.

We have added text to the main manuscript describing the bedrock source within each moraine (lines 85-93).

Finally, a recent paper on the Mount Archenar moraine states that “ the similarity to Lake Untersee suggests commonality between Mt. Achenar Moraine with other Antarctic sites where brine formation is minimal.” (Graly et al., 2020). Could this be discussed?

Graly et al., 2020 study subglacial chemical weathering beneath the Mount Achenar by analyzing sediments at the very edge of Law glacier. They describe chemical weathering in an aqueous environment that is generally consistent with our view of calcite endmember waters. Their chemical weathering reactions include oxidation of carbon from Beacon supergroup and Ferrar dolerites, where carbon is the dominant anion in the system. These oxidation reactions are made possible in the closed subglacial system by oxygen delivered to the system vial glacial meltwater. Cations are delivered to the subglacial system via leaching of comminuted ground mass.

However, in these marginal areas of Law Glacier, Graly infers short water residence times (on the order of 60 yr), which is far shorter than our >10 kyr record implies. Graly also has no spatiotemporal information about these sediments, so it is unclear what part they could play in the history of the subglacial precipitates discussed in our manuscript. While Graly states that the chemical compositions they demonstrate at Mt. Achenar are generally consistent with aqueous systems where brine formation is minimal, they also mention that the cation composition is within the range of Blood Falls, a subglacial brine.

Graly draws a comparison with Lake Untersee waters because they share carbon as the dominant anion, and this area of Mt. Achenar has water sourced primarily from glacial meltwater. They also have cations sourced from an igneous substrate with little sulfur and carbonate. However, beyond these very generic similarities, the chemical characteristics of the water that resulted in the precipitation of PRR50589 and MA113 are quite different. For example, because the samples form from 2 different waters that mix, we are able to identify that one water is extremely carbon poor and highly saline, which are very different waters than those inferred by Graly or measured at Untersee. Another key difference between subglacial conditions and those observed at Lake Untersee, is that Untersee receives sunlight and has photosynthetic organisms. It is likely much less isolated than the aqueous systems we describe, as it could have been in contact with the atmosphere as little

as 500 yrs. ago (Marsh, 2020), while the $\delta^{13}\text{C}$ compositions of our samples require that they formed incomplete isolation from the atmosphere isotopically heavier atmosphere.

2) Petrographic considerations

Petrography is entirely missing, apart from images of polished slabs. Here are some points relevant to the results and discussion:

a) From Extended Data Fig. 1, the PRR50489 and MA133 it appears that they have different petrographic characteristics. MA133 looks more similar to a true subglacial precipitate and PRR50489 looks more like what Faure et al's (1988) inferred it to be: a subglacial travertine dislocated by subglacial tectonics.

We understand the reviewer has an expertise in petrography and agree that petrographic consideration would be an interesting follow-up way of characterizing these precipitates. However, the paper presents age and isotopic constraints that provide strong evidence that suggests that these samples are both subglacial precipitates that form through similar processes; the collective dataset does not support the role of volcanic fluids. Below we list these constraints and refer to them in response to similar comments later in the review.

The similarities in MA113 and PRR50489 include:

- 1) Both samples consist of opal and calcite layering that reflects a cyclic change in subglacial water compositions.*
- 2) The timescales of transition between opal and calcite in both samples occurs on millennial timescales.*
- 3) Opal and calcite layers are compositionally distinct with respect to:*
 - a) $^{87}\text{Sr}/^{86}\text{Sr}$, requiring that the opal and calcite forming waters experienced a distinct chemical weathering history and contacted compositionally distinct bedrock.*
 - b) $\delta^{13}\text{C}$, requiring distinct sources of carbon for each water.*
 - c) $\delta^{18}\text{O}$, requiring different sources of glacial ice for each water.*
 - d) Ce^* , implicating different redox conditions for opal and calcite waters*
 - e) Sr concentration, implicating the Sr rich opal forming water as higher salinity water, while the calcite endmember water is relatively dilute.*
 - f) C concentration, implicating the C-poor opal forming water from C poor brine, while the calcite water is carbon rich.*

The fact that these characteristics listed are observed in both samples, strongly suggests that these samples formed through the same processes. A coincidence seems unlikely, particularly given that these compositional traits match that of actual waters found beneath the ice sheet. As such, we respectfully disagree that MA113 and PRR50489 form through different processes.

Hydrothermal fluids: While no single variable in our geochemical dataset unequivocally rule out the presence of hydrothermal water, we argue that our collective dataset can be simply explained through normal glacial processes and with fluids readily observed

beneath ice sheets. While calling on a hydrothermal fluid endmember would require more assumptions and an overall more complex explanation. Below we outline specific isotopic data that would be unlikely to occur in the presence of hydrothermal fluid. We refer to these data in later comments discussing the possibility of a hydrothermal fluid source.

- 1) The $\delta^{13}\text{C}$ of the calcite endmember water for both samples clusters tightly at a highly negative value (ca. -23‰ for PRR50489 and ca. -18‰ for MA113). Water with a uniform $\delta^{13}\text{C}$ that is this ^{13}C -depleted would be unlikely in Faure's proposed scenario where carbon is sourced to subglacial aqueous systems when hydrothermal waters contact the substrate. Although, like Faure points out, Beacon sedimentary rocks include coal with $\delta^{13}\text{C}$ values close to PRR50489, recent measurements of bulk $\delta^{13}\text{C}$ in soils in this area of the TAM describe bulk carbon values that are much more ^{13}C -enriched (0.2 – 8.5‰; Diaz, et al., 2020). In Faure's scenario, this enriched carbon would mix with carbon sourced from coal and would result in water with an intermediate $\delta^{13}\text{C}$ value. Instead, uniform $\delta^{13}\text{C}$ compositions clustering around the value of terrestrial organic carbon indicates that microbes in the aqueous system selectively oxidized the most readily available carbon source (likely fossil terrestrial organic matter; we explain this further in a comment below), rather than the carbon that would be included by a hydrothermal source. Such uniform $\delta^{13}\text{C}$ values have been found in sediment-rich basal ice beneath the AIS (Montross, et al 2014; Yan, et al 2019), and have been a hypothesized process in subglacial aqueous environments (Graly, et al., 2017; Marsh, et al., 2019).
- 2) The $\delta^{18}\text{O}$ of all precipitates we've measured match the oxygen composition of glacial meltwater at 0°C. Unlike the opal $\delta^{18}\text{O}$ value from Faure et al., 1988 (which we were not able to reproduce; Table S3), MA113 and PRR50489 opals also have $\delta^{18}\text{O}$ compositions that match glacial water.

In Faure's explanation for the $\delta^{18}\text{O}$ values, subglacial hotspots drive spring water to the surface, which melts the overlying ice and imparts oxygen compositions with a glacial value. As a first order test of this mechanism, we consider a mass balance to simulate the mixing proportion of glacial water and hydrothermal water necessary to achieve the $\delta^{18}\text{O}$ value of our calcite endmember water from PRR50489 (Fig. 4a):

$$-56‰ * X + 0‰ * (100 - X) = -55.3‰ * 100$$

Where the $\delta^{18}\text{O}$ of basal meltwater is set to -56‰ based on the most ^{18}O -depleted value in the Dome-C ice core (this is a conservative estimate, as the basal ice value proximal to Elephant moraine is very likely more ^{18}O -depleted). The hydrothermal endmember is estimated to be 0‰ (which is also conservative, as hydrothermal fluids have $\delta^{18}\text{O}$ that are readily well above 0‰). Even in this very conservative scenario, to achieve the calcite-endmember composition from PRR50489 (-55.3‰) would require that 98.75% of the water in the subglacial system was derived from the melting of glacial ice. However, meteoric ice is dominated by marine Sr and U isotopic compositions (Aciego, et al., 2011), thus this scenario could not explain the high $^{234}\text{U}/^{238}\text{U}$ and $^{87}\text{Sr}/^{86}\text{Sr}$ observed in our samples, which instead, require long-term water-rock

interaction. What is more, the $\delta^{18}\text{O}$ value of the calcite endmember water from MA113 is -61.2‰, which is more depleted than any ice upstream of Law Glacier (South Pole Ice Core: Steig, et al., 2021). It is likely that this extremely low $\delta^{18}\text{O}$ value results from some portion of the calcite-endmember water experiencing freezing prior to mixing to form the calcite layer. For this $\delta^{18}\text{O}$ value to result from Faure's hydrothermal mechanism would require basal ice that is even more depleted than -61.2‰, which is highly unlikely given that it has never been observed in an Antarctic ice core.

- 3) *Elevated $^{234}\text{U}/^{238}\text{U}$: Enrichment in ^{234}U is consistent with a surface water and would be rare for a hydrothermal water. The extreme (500%) ^{234}U enrichment observed in our precipitates occurs from prolonged (>10 ka) water-rock interaction at the ice sheet-substrate interface. While hydrothermal water can achieve this composition following prolonged leaching of bedrock, mixing with melted glacial ice would dilute this signal with a $^{234}\text{U}/^{238}\text{U}$ signal of 1.14 (Aciego, et al 2011). Subglacial waters with elevated $^{234}\text{U}/^{238}\text{U}$ are not rare in Antarctica. Similar compositions are observed in Blood Falls brine (Lyons et al., 2020), Don Juan Pond (Henderson et al., 2006), and Antarctic Precipitates found at Lewis Cliff Glacier (Fitzpatrick et al, 1992), the Pensacola Mtns. (Blackburn et al., 2020), Law Dome (Goodwin et al., 2017) and Boggs Valley (Frisia et al., 2017). Perhaps most importantly, precipitates from the Laurentide ice sheet (Refsnider et al., 2014) and Yosemite Valley (Blackburn et al., 2020), areas that are not volcanically active, also record ^{234}U enrichment. This prevalence of ^{234}U enrichment in subglacial waters across a wide range of space and time is an indication that long-term residence of water beneath ice sheets is a common occurrence.*

If hydrothermal waters were present, they would likely affect all of these isotopic compositions. Further, hydrothermal waters have been called on as a source of subglacial melting (Frisia, et al., 2017). However, recent models provide evidence that much of the base of the EAIS in its interior is at or close to the pressure melting point (Pattyn et al., 2013; Van Leifferinge, et al., 2017). We provide a glaciologic explanation and model (shear heating driven by ice sheet velocity changes) for the episodic introduction of meltwaters to ice sheet margins. Therefore, we do not identify a need for a subglacial hydrothermal melting source for precipitate formation.

In summary, our geochemical observations from precipitates MA113 and PRR50489 provide evidence that opal and calcite layers formed from mixing of two different fluids, glacial meltwater and brine. These fluids have compositions that match waters readily found beneath ice sheets, even in volcanically inactive areas. We view our explanation for formation via introduction of meltwater to a marginal brine system as the most parsimonious process to explain our samples.

Also, MA133 appears to have two different facies, one consisting of brown, coarse calcite and opal lenses and one consisting of darker, fine grained (?) calcite and opal. This may signify a major change in the environment or a different provenance of the fluids...also for the calcites.

The top ~1.5cm of MA113 consists of mostly of silicate detritus with calcite cement, and one opal layer. We agree that this likely represents a large change in the subglacial

environment, either a disruption of the subglacial cavity and/or large fluid events. Because this part of the samples is not uninterrupted calcite and opal layers, we cannot interpret these two-rock types in the climatological framework, and therefore do not include it in the spectra in figure 1 and view it as beyond the scope of this project. To make this clearer, we have labelled the unconformity in the figure, added to the figure caption that all data presented are from below this unconformity, and add lines to make it clear where spectral data are taken from.

PRR50489, I believe, should come from the suite of samples collected by Faure and colleagues in their explorations. On the basis of Sr, O and C isotope ratios and trace element concentrations Faure et al. (1988), proposed that the Elephant Moraine subglacial calcite-opal deposit built up in “hot-spots” that gave rise to subglacial springs, where travertine-like deposits formed. The EM sample illustrated in Faure et al (1988) Figure 2 (Sample 582) looks very much like a well-laminated spring deposit. Similarly, PRR 50489 “aspect” is more similar to that of a travertine than “classical” subglacial calcites, with their fluted and relatively delicate morphologies.

Faure’s interpretation of PRR50489 as a “subglacial travertine”, was made before the discovery of widespread subglacial meltwater and brines. Without the knowledge of old, isolated sub-AIS groundwaters, Faure would have logically assumed that warm groundwater springs are required for water to be present and were the simplest explanation for abiogenic opal precipitation.

The interpretations of Faure, et al 1988, of high precipitation temperatures were also inferred to explain $\delta^{18}\text{O}$ measurements. However, our seven opal $\delta^{18}\text{O}$ measurements from PRR50489, and three opal $\delta^{18}\text{O}$ measurements from MA113, demonstrate water values that are >10‰ more depleted than Faure’s. Although not included in the original dataset, we remeasured opal from precipitates EM-2, which Faure measured in 1988 (now included in table S3). Contrary to Faure’s $\delta^{18}\text{O}_{\text{OPAL}}$ measurement of +13‰, we measure $\delta^{18}\text{O}_{\text{OPAL}}$ value of +2.86‰. Instead of a warm groundwater spring as the source for the opal endmember, this $\delta^{18}\text{O}$ measurements is consistent with opal formation from water derived from glacial ice, with a composition of -40.5‰ at 0°C.

Faure argued that waters that precipitated the Elephant Moraine precipitates were of glacial origin, but they would not have known that large portions of the EAIS bed are at the pressure melting point, which would have alleviated their need to call on a heat source to melt glacial ice. The $^{87}\text{Sr}/^{87}\text{Sr}$ presented by Faure (which generally match values we’ve produced here) can be explained by protracted exposure of subglacial water to underlying bedrock/sediments, which again Faure would not have been aware of.

While we acknowledge that the exact source location for PRR50489 is unknown, as described above, we find no isotopic, redox, or geochemical evidence that would suggest a travertine or hot spring origin.

Fluted morphologies characterize two other specimens of Antarctic subglacial carbonates (aragonites mostly) illustrated by Aharon (1988), which were sampled on gneissic rocks of the Vestfold Hills and the Boggs Valley calcites, which were also collected in-situ on amphibolites.

Both these two examples of Antarctica subglacial carbonates are different petrographically and mineralogically from the PRR 50489 moraine sample. The MA 133 seems to show sparite and microsparite like the Boggs Valley samples in its lighter part, but then it becomes different (Extended Data Fig.1, panel c).

As neither PRR50489 nor MA 133 were “in-situ”, it becomes hard to tell if they were ice-flow oriented subglacial deposits formed from a film of fluid confined by regelation ice (*sensu* Hallet, 1976; Siman-Tov et al., 2021) or not. Siman-Tov et al., (2021) provide a key for the development of similar subglacial deposit, which I believe is the hypothesis embraced in the present manuscript.

We acknowledge that Hallet and Siman-Tov provide a useful framework for understanding subglacial precipitate formation. However, they describe precipitates forming in alpine regions that are not necessarily analogous to all Antarctic catchments. PRR50489 and MA113, like many other subglacial precipitates observed forming in valleys on the EAIS side of the Transantarctic Mountains, show characteristics of formation in larger subglacial cavities. PRR50489 and MA113 are 3 and 9 cm respectively, which strongly refutes precipitation within a basal film, and instead suggests formation from fluid in a subglacial cavity. They also show signs of variable precipitation mechanisms between calcite and opal, where calcites seem to form relatively quickly on the substrate and opals form via settling from the water column. Together these observations rule out formation via a film, but also do not fit the canonical regelation formation mechanism. We have added this context to the main text (lines 107-110).

This mechanism of subglacial deposits formation implies one type of water flowing through interconnected subglacial conduits (which formed the calcite) and the refreezing of subglacial meltwater occurred and formed opal. The transport of water through conduits transports, however, should carry lithic fragments and fragments of broken subglacial deposits itself, which then would settle from the fluid and become entrapped in the regelation step (Frisia et al., 2017; Siman-Tov et al., 2021). It then stands to reason that at the boundary between the calcite and the opal (transition from open to closed system), the detritus should settle, particularly in furrows. However, this is not commonly observable in the samples. Detritus should also be present at the top of the opal, marking the onset of flow in the open system. The hypothesis that open system fluids form calcite and refreezing of basal meltwater form opal should petrographically result into: i) formation of microsparite at the lower contact between opal and calcite (cf. Sharp et al., 1990; Frisia et al., 2017); ii) embedding of detritus in the microsparite; iii) re-nucleation of calcite above the opal followed by geometric selection. These petrographic observations would provide proof of concept.

This is an interesting set of hypotheses with which we can approach textural analyses of these samples. One thing to note, though, is that there are several unknown physical parameters about the aqueous system that may complicate these three textures. First, we do not know the velocity of waters flushing during periods of subglacial connectivity. Since we call on basal shear heating as a mechanism to open basal conduits, these events need not be catastrophic floods like those observed in active, marginal, subglacial hydrologic systems. Secondly, we do not know the amount of physically weathered detritus in the subglacial environments beneath the EAIS, and it is likely that there is less micro-detritus

in these environments than in more marginal systems (Jameison, et al., 2010) where ice flows more rapidly. Finally, we do not know the depth nor horizontal extent of the subglacial basin. Therefore, sediment delivery could be concentrated near the mouth of the basin but would not necessarily be basin wide. With these caveats in mind, we agree that these textures could provide a proof of concept but argue that their absence does not nullify our conclusions, which we base on chemical data.

Nevertheless, we have preliminary maps and hand-sample scale observations that seem to match some of the proposed textures. In MA113, detritus is concentrated in calcite layers, and in many cases, sand-sized grains clearly sit atop flat opal layers. Based on their uniform white textures, opal layers in both samples are exceptionally free of clasts larger than silt sized, suggesting deposition in an energetically quiet environment. XRF element maps images along the opal-calcite layer boundaries of PRR50489 display evidence of rip-up clasts from the underlying opal, and detritus. We have added mention of these textural analyses as well as relevant data to the supplementary text and in supplementary figures S1-S3.

Microsparite seem to occur in the MA 133 polished slab illustrated in Fig 1 of the Extended Data, but there are two very distinct upper opal lenses that appear to be in direct contact with sparite at the bottom.

We find these dirty microsparite layers at the top of the sample intriguing as well. They clearly represent a disruption to the subglacial system that could be a more energetic flushing event or a partial closure of the subglacial catchment. Based on U-series ages we can bracket this dirty microsparite layer between 23 kyr and 42 kyr, which raises the possibility that this layer is the result of a large flushing event leading up to termination 1. The distinct opal layer within this layer could have occurred during an ensuing millennial-scale cold period. Unfortunately, we do not have the age resolution to precisely date this part of the sample, and we therefore do not include it in the manuscript.

I use the term “lenses” because the opal layers commonly “taper” out, which is very clear from Fig. 1 in the Extended Data.

The bottom of the opal lenses is almost always concave and the top is flat (even in the PRR sample, where they may be, however, deformed by glaciotectonics). Similar morphologies are hard to find in the literature of subglacial carbonates where I found visual illustrations of the petrography (Bjaerke and Dypvik, 1977; Aharon, 1988; Sharp et al., 1989; Refsnider et al., 2014; Lipar et al., 2021). The only one I could find is Siman-Tov et al. (2021), where thin sections and chemical maps show Si-rich (detritus-rich) layers and calcite layers having flat contacts both at the bottom and the top of the layers. But not a concave bottom and a flat top and detrital-rich layers are common, whilst similar detrital-rich layer are impossible to discern from the slabs in Fig. 1 ED.

Now, the concave lower boundary of opal results from the occurrence of calcite botryoids. The opal may even taper out against calcite. Now, I am wondering if that calcite is heteropic with the opal or not. It would be worth dating.

We agree that subglacial precipitates with opal-calcite transitions are not described in the literature. While Siman-Tov describes precipitates that record changes in subglacial water composition, we do not view this precipitate formation mechanism as analogous to Antarctic subglacial environments. Our precipitates require a subglacial cavity that stays open on 10kyr timescales, and records periods of both isolation and connection to the broader subglacial water system. Opal does, indeed, fill depressions made by calcite botryoids in layers with flat tops. We interpret this texture as evidence for opal formation in an energetically quiet environment, where high concentrations of Si drive opal out of solution in the water column. This colloidal opal settles out of the water column, filling the depressions of the substrate. The flat top of these opal layers is a clear indication that these laminae are gravity lain. We've added text to the main manuscript to make this clearer..

Based on our U-series ages versus the layer depth in the sample, and the inferred formation mechanisms, we expect that opal formed slowly and calcite formed rapidly. Unfortunately, we cannot date the calcite layers because their U/Th is too low to measure precise U-series ages.

The visible stratigraphy suggests that opal deposited in furrows between calcite botryoids, rather than in distinct layers. And there is no evidence that the crystal tips of the calcite are corroded. The Si synchrotron map, despite the large pixels, is good enough to suggest that the calcite underlying opal has pristine crystal terminations (PRR50489). Now, would it be a possible alternative explanation that calcite precipitation continued at the tip of botryoids exhausting most of the Ca in solution and furrows hosted micro-reducing systems because of the complete exhaustion of O by subglacial microbial metabolism?

We agree that opal does fill furrows in the calcite, but opal precipitation appears to continue for thousands of years in some cases, and in most cases, builds mm-to-cm thick layers that exceed the height of calcite crystals.

While we appreciate the thought this reviewer demonstrates in offering this alternative hypothesis, we do not feel that the combined textural evidence in the samples, paired with chemical and chronologic evidence, supports opal-calcite transitions from chemical transitions (i.e. redox changes) in a single fluid. We agree with the observations that the tips of calcite crystals are not corroded. However, based on Ce data the opals exhibit varying redox conditions between oxidizing (low Ce* values) and more reducing (high Ce* values). This range of Ce* values correlate well with how much of the brine endmember component (Fig. 2 color-scale), which we interpret as strong evidence for mixing of two waters, rather than a redox controlled process. Additionally, we do not see any evidence for microbially mediate precipitation in either sample. We would expect a similar isotopic composition for both calcite and opals if they were forming from one water, but based on individual measurements, and made clearer from mixing models, we show a clear difference in $\delta^{13}\text{C}$, $\delta^{18}\text{O}$, and $^{87}\text{Sr}/^{86}\text{Sr}$ between calcites and opals.*

We do admit that with redox proxies we present, we do not have strong evidence that opal forming waters were fully reducing, rather than intermediate waters. We have softened

language to allow that opal forming waters were anywhere between intermediate redox conditions, and fully reducing.

I was wondering this also because there are some opal layers with no Ce* anomalies and some calcite layers with Ce* anomalies (fig. 4, Extended data)?

As mentioned previously, we find the correlation between Ce and position on isotopic mixing models (Fig. 2) as compelling evidence for varying mixing ratios of two chemically distinct fluids.*

Maybe the system shifted laterally, as it is possible in the complex topographic substrate of the subglacial environment, rather than shifting through time. This does not contradict the hypothesis of externally forced subglacial system changes, it would put it in its environmental perspective, where lateral changes are common (see, for example, the lateral variability of coeval subglacial samples illustrated by Frisia et al., 2017)

We presented a mechanism for opal-calcite precipitate transitions that is a lateral shift in subglacial conditions on a regional scale driven by ice velocity changes in the region. Based on the variable $\delta^{18}O$ and $\delta^{13}C$ values between the two endmember waters, we expect that the lateral transition that drives connectivity between the two waters need connect interior waters with marginal systems. This mechanism is reasonably similar to the drainage described by Frisia et al., 2017, save for the hydrothermal influence on the upstream lake.

We acknowledge that the basal thermal regime can be highly complex under AIS outlet glaciers and can have local shifts between frozen and unfrozen conditions. We address this point in the paragraph starting on line 423 in the original main manuscript (paragraph now starting on line 341 in the revised manuscript). While lateral shifts in the subglacial system are entirely possible, we argue that the correlation between the opal-calcite transitions and SH climate cycles over the ~100kyr in two distinct areas requires a highly regular triggering mechanism. The chemical compositions of the two endmember waters corroborate this hypothesis, because they demonstrate that an interior meltwater, and marginal brine are ideal candidates for forming calcite and opal respectively.

The teleconnection hypothesis could better support the existing data particularly given that only one sample has good ages and, overall, the whole interpretation is based on one sample.

We admit that part of sample MA113 posed difficulty for dating, but the isotopic and geochemical data suggest an identical formation to that of PRR50489. The correlation between the millennial climate-cyclicity and the MA113 opal-calcite transitions are very good for the part of the record that is well dated. The part of the record that does not correlate well (i.e., parts of 49 – 53ka) occurs when Southern Hemisphere climate did not experience strong millennial cycles. We propose that at this portion of the record, the climate was close to the threshold between opal and calcite precipitating conditions. In the revised manuscript, we describe opal-calcite transitions as the result of ice motion in response to reaching climate thresholds, beyond which the ice sheet experiences dynamic

change. We investigate the probability that opal-calcite transitions occur following ice motion in response to passing a climate threshold and find that a significant portion of the mineralogic record matches this interpretation (paragraph starting on line 289).

While we hope to expand on climate record presented by subglacial precipitate archives in the future, we want to point out that the 100kyr climate record in PRR50489 and the 12kyr record in MA113 are analogous to speleothem records. That is, although these are two samples from two locations, they represent a large range in time, that can be viewed in a similar light as other climate archives from separated locations.

b) Another important point for petrography to be carried out is from Faure et al's (1993). In discussing the high Sr isotope ratios in their non-marine carbonates they suggested possible contamination with fine detritus (clay). I am wondering if the presence of nano- sub-micrometre sized particulate could be an issue also in the two samples analysed here. Could particulate contained in the calcites not be picked up by EDS and have contributed to their Sr isotope ratios? Note that the sample from the Elephant Moraine analysed by Faure et al (1998) has high $^{87}\text{Sr}/^{86}\text{Sr}$ isotope ratios, which are in the range of the samples measured in the present research and in the range of minerals formed from fluids that came into contact with the Beacon Supergroup of the Central Transantarctic Mountains (cf. Faure et al., 1993). This is why I believe that a geologic contextualization is worth looking into.

While a small amount of detrital affect is possible in $^{87}\text{Sr}/^{86}\text{Sr}$ values, we do not observe any evidence for substantial micro-silicate clay components in EDS or XRF maps for either sample. Sample MA113 does have some silicate detritus in its calcite layers, but we are able to avoid that when sampling for Sr isotopic analyses. If, as the reviewer suggests, clay particles were well below our $1\ \mu\text{m}$ pixel size for EDS maps, we would still expect them to affect the color of the pixel in Al or Si maps (one $1\ \mu\text{m}$ pixel in that case being a mix of primary precipitate material and clay particles), which should still be visible if they made up a significant portion of the rocks or were present in varying amounts in calcite layers. However, we do not observe this in our analyses of calcite. In the event that silicate detritus was included in calcite samples, they would dissolve during the preparation of calcite samples for Sr measurements, and, therefore, would have a negligible effect on the measured Sr compositions. While it is more difficult to observe detritus in opal maps, the tight range in opal $^{87}\text{Sr}/^{86}\text{Sr}$ values, and their white color suggests that these are free of a detrital component.

We agree with Faure's interpretation that the Sr compositions of these precipitates are sourced from crustal rocks. However, we find it a more parsimonious explanation that these Sr values are imparted on waters during long term water-rock interaction, rather than from clay components. The measured Sr isotopic compositions do, indeed, match those of Beacon supergroup or Ferrar dolerite (which themselves have a very wide range on Sr composition). As such, we have made note in the main text of Faure's observation of Beacon supergroup and Ferrar dolerite at elephant moraine. However, the calcite endmember $^{87}\text{Sr}/^{86}\text{Sr}$ values are substantially lower than opals and plot over a range of compositions. Rather than resulting from variable amounts of microparticles, we hypothesize that this large range in calcite $^{87}\text{Sr}/^{86}\text{Sr}$ implies mixing of two waters with

distinct compositions. Within this framework, the opal waters dominate the Sr composition because they contain much higher Sr concentrations, and calcite Sr compositions depend on the mixing ratio between the two end members. This explanation fits well the collection of geochemical evidence we present for a brine and meltwater endmember.

3) Diagenetic considerations

It is most likely that opal had a precursor. I suppose that opal in the PRR and MA samples is now Opal CT, thus, most likely there was a transformation from Opal A to Opal CT and, perhaps, from a polymerized colloidal suspension to Opal A to Opal CT as it happens in warm springs or in discrete silicification of cave carbonates. The hydration state of the constituents of opal and Ostwald ripening may have a role in kinetic uptake of elements and isotope fractionation

Although not presented in the original article, we have collected XRD data opal layers that demonstrate that opal in these samples is opal A, and therefore experiences minimal diagenetic alteration. We've added these data as supplementary figure S3 .

There are kinetic effects associated with dehydration and rehydration during the transformation. REE, for example, are preferentially uptaken in the hydrated phases within a general reducing environment. Kim et al (2012), for example, suggested that negative Ce anomalies might not always be an indicator of paleo-oxic conditions. For REE fractionation, a positive Ce anomaly may appear in particulate organic material and negative Ce anomaly may result from Ce scavenging in marine examples (Haley et al., 2004). Thus, may it be possible that the Ce* is localized and that also some calcite may have positive Ce*? This potential role of diagenesis should be considered in the discussion and either accepted as complexity or discarded on some other basis.

We agree that there are potential complications to applying paleoredox proxies, like Ce. The idea presented by Haley et al., 2004 – where negative Ce anomalies in oxygen-rich marine waters result from scavenging of insoluble Ce(IV) – is the same mechanism shown to drive Ce-anomalies in most waters (Tostevin, et al., 2016), is the mechanism we embrace for negative Ce* of calcite, and does not represent a potential disruption of Ce* as a paleoredox proxy. However, the idea posed by Kim et. al 2012 – that negative Ce* in pore waters can be induced by microbial scavenging of Ce⁴⁺ in reducing fluids – is an intriguing possibility that we discuss below. We note that this mechanism does not apply to positive Ce* values, like those we observe in opals, which are caused by the dissolution of Ce-bearing phases. Therefore, we focus this response specifically on the idea that calcites could have negative Ce* in reducing fluids.*

The Kim et al., 2012 paper describes the disruption of Ce as a redox proxy specifically when negative Ce* values develop in reduced waters, which occurs via Ce scavenging by organic material. This mechanism was first described experimentally by Pourret, and others in 2008, who demonstrate Ce scavenging by humic acid in highly alkaline waters leading to negative Ce anomalies in the absence of oxygen. Due to the lack of oxygen, their results require an alternative mechanism to oxidize Ce(III) to Ce(IV). They call on oxidization by dissolved carbonate in solution, which means this mechanism is applicable*

only in highly alkaline waters, where carbonate ion is dominant carbon species and is highly concentrated.

The required conditions described by Pourret run contrary to many of the characteristics we describe geochemically in the subglacial waters where our precipitates form. First, the $\delta^{13}\text{C}_{\text{ORG}}$ in calcite requires that organic material is sourced to provide carbon to the calcite endmember fluid. If organics were scavenging Ce, this would lead to positive or neutral Ce anomalies in the calcites. Next HCO_3^- is the dominant anion in most subglacial fluids that have been probed; the difference between $\delta^{13}\text{C}_{\text{ORG}}$ and $\delta^{13}\text{C}_{\text{DIC}}$ match the HCO_3^- - CaCO_3 fractionation suggesting that the carbon in the system is dominantly HCO_3^- . It is unlikely that these subglacial fluids would have high enough pH and alkalinity to contain high concentrations of CO_3 .

It would also be hard to explain positive opal Ce values if the scavenging material was organics. Mixing models require opal solutions to contain <3% of the carbon in the system. If carbon was mixed with high cation opal endmember (assumed based on Si concentration) this would most likely lead to calcite precipitation, thus violating the monomineralic opal we observe. It is a much simpler explanation to have Ce scavenged by an oxide that is then dissolved when solution gets more reducing (i.e. the mechanism described by Tostevin, et al. 2016), which would be expected during cryoconcentration/isolation.*

Diagenesis may impact other isotope/element exchanges, particularly when the system becomes open again. Mixing of fluids of diverse ionic strengths, particularly brines, is known to cause dissolution and subsequently precipitation of carbonates.

We cannot rule out complete dissolution and reprecipitation of calcite layers during the ~1000 yr AIM period. We mention missing time in calcite layers is possible but would not affect the chronologic correlation between mineral transitions and millennial-scale climate cycles.

However, the chemical characterization of calcite layers, in our view, seems to align with primary signals from a distinct subglacial fluid. First, the $\delta^{18}\text{O}$ values of calcite are systematically lower than the brine endmember. If calcite was dissolved and reprecipitated in the brine, the two fluids would have similar oxygen isotopic compositions. Similarly, the uniform $\delta^{13}\text{C}$ composition of calcite layers that matches oxidation of organic carbon suggests precipitation in the absence of the brine endmember. Our mixing models (Fig. 2) suggest that calcite compositions are representative of 80 and 95% glacial meltwater endmember, with variable amounts of brine. Further, the fibrous appearance of calcite layers suggests at least minimal dissolution of these particular layers. We add of these analyses in main text (line 114-117).

Sample MA 133 is characterized by large uncertainty in the ages (Fig. 2 in Extended data) that are of the order of thousands of years. A few ages even suggest the possibility of reversals. In Fig. 1 main text, the whole uncertainty of the ages should be reported, not the mean point, to

show how a shift of a few thousands of years to the right or left of the mean point would not match the WDC delta18O normalized dataset.

The low uranium concentration of MA113 created difficulties in dating. However, there is no a priori reason to call on an age reversal, given that the ages are within uncertainty. We argue that the macro-scale textures of the samples including fibrous calcite crystals and draped opal layers allow for the assumption of stratigraphic consistence.

Bayesian modeling of age models using stratigraphy a prior is now a commonly used method (e.g. Deino, et al 2019; Schoene, et al., 2019). We describe this, and the limitations of this approach in the methods section of the original manuscript.

We have added modeled uncertainties to Fig. 2 and 3 and refer to them when describing the limits of interpreting the correlation between the climate records and our sample mineralogy.

The correlation is, in my opinion, a bit stretched even assuming “stratigraphic consistency” (Methods). MA133 dataset potentially point to a diagenetic history (which may be also valid also for PRR50489, but with minor impact).

As was mentioned in the previous response, the assumption of stratigraphic consistency is used only in the Bayesian model as a prior to improve measurement precision.

We admit that, within dating uncertainties, we cannot assess sub-millennial differences between our model and climate. In the context of this correlation, any diagenesis likely falls within this dating uncertainty. However, there is a clear correlation between the opal-calcite records and the ice core climate proxy records, save for the early part of the MA113 record. This visual consistency is remarkable given that the opal-calcite mineralization the manifestation of several ocean-atmosphere-cryosphere teleconnections many of which behave non-linearly. Given that one should not expect subglacial environment to vary perfectly linearly with climate, we have added an experiment that explored opal-calcite mineralization in response to the crossing of a climate threshold. We describe this experiment in the paragraph starting on line 289, and in various parts of the discussion section.

There is a potential for diagenesis for both Opal (following Ostwald’s step rule) and calcite, due to the mixing of the fluids (brine and subglacial meltwater) with different ionic strengths. It would be rather interesting to see what the tip of the calcite crystals really look like in thin sections, and what the top of the opal layers also looks like, where the diverse fluids mix.

Thank you for this suggestion. We agree that looking in detail at the microtextures within these precipitates could yield interesting results for understanding the evolution of subglacial meltwaters. While we view this effort as beyond the scope of the reported manuscript, we plan to continue these types of analyses within later studies.

4) Some implications of points 1-3 for Isotope ratios data and trace elements

Faure et al (1998) implied that the underlying bedrock of the EM carbonates is the Beacon Supergroup, because of the depleted $\delta^{13}\text{C}$ compositions (-23‰) of the calcites. These were explained as due to the hydrothermal fluid (in the present submission, hydrothermalism is ascribed to reference 45, which is actually cited erroneously and out of context as I will explain later. The correct reference here is Faure et al., 1998)

Thank you for pointing out this erroneous reference, we have made the suggested change.

We note that we cannot locate the suggested reference: Faure, 1998. Perhaps the reviewer refers to: Faure G (1998) Principles and applications of geochemistry, 2nd edn? We instead cite Faure 1988 where applicable, and in other cases cite Faure and Mensing, (2011). The Transantarctic Mountains.

that came into contact with the coal, in analogy to other measured $\delta^{13}\text{C}$ values of calcites associated with coal. If this is the case, the C of the PRR 50489 would not be sourced by subglacial organic matter in a strict sense (which should be organic compounds/microbes) but from Permo-Triassic coal.

We agree with Faure's original interpretation that fossil carbon from ancient sediments is a likely carbon source, however we reinterpret the process by which this carbon is introduced to subglacial fluids.

We find it likely that hydrothermal fluids would source carbon both from organic material but also from the sedimentary rocks they contact, which would impart a heavy $\delta^{13}\text{C}$ signal that would mix with the depleted carbon signal in a larger proportion than we observe in the calcites. In characterizing rocks from Elephant moraine, Faure notes abundant "calcite cleats" and crustal rocks from the Beacon supergroup and Ferrar dolerite, which would add a heavy $\delta^{13}\text{C}_{\text{DIC}}$ signal. Yet, all calcite values in PRR50489 cluster around -23‰, suggesting that the vast majority of carbon is sourced from organic matter. One caveat to this interpretation is if the hydrothermal fluids mixed heavy $\delta^{13}\text{C}$ materials with an extremely depleted material, like methane. However, as we will describe in the following paragraph, we do not observe any evidence for methane in our samples.

Since Faure's publications calling for hydrothermal processes, several sub-AIS waters have been probed and a large, microbially mediated, subglacial carbon reservoir has been revealed. The $\delta^{13}\text{C}_{\text{ORG}}$ values of these waters are uniformly around the composition of terrestrial organic matter. While not in the original manuscript, we have measured $\delta^{13}\text{C}_{\text{ORG}}$ within both PRR50489 calcites and find their $\delta^{13}\text{C}_{\text{ORG}}$ values cluster around -26‰, indicating that, like other observed sub-AIS lakes, these aqueous systems have uniform $\delta^{13}\text{C}_{\text{ORG}}$ likely associated with microbial respiration of terrestrial organic carbon. We have included these $\delta^{13}\text{C}_{\text{ORG}}$ measurements to the supplementary data table and have added text to the main manuscript that describes how these data aid in our interpretation of microbial respiration of terrestrial organic carbon as the main carbon source (lines 159-168).

It is a tiny difference, but it matters. Microbial oxidation of subglacial organic matter of microbial origin would yield more positive values, around -12‰ (Mook, 2006).

In most systems on the surface of the Earth, calcite would, indeed, have ~12‰ more positive values than the $\delta^{13}\text{C}_{\text{ORG}}$ of the system. However, for aqueous systems that remain closed to the atmosphere and are within silicate terranes, the $\delta^{13}\text{C}_{\text{DIC}}$ can be close to the $\delta^{13}\text{C}_{\text{ORG}}$, as we observe in PRR50489 (Clark and Fritz, 1997; Barth, et al., 2003; Graly, et al., 2017). In the case of closed systems, microbial respiration can function as the only significant source of CO_2 . This CO_2 undergoes hydrolysis to H_2CO_3 and is fully utilized in the chemical weathering of the substrate. In closed systems where there is no carbonate to be dissolved, all the HCO_3^- in the system will be sourced from silicate weathering, which would result in $\delta^{13}\text{C}_{\text{DIC}}$ equal to $\delta^{13}\text{C}_{\text{ORG}}$. This process was first described by Clark and Fritz in 1997 and was shown to affect $\delta^{13}\text{C}_{\text{DIC}}$ of river waters by Barth and others in 2003; Graly and others described this as a process that could affect subglacial systems in a paper published in 2017. We find this to be a compelling explanation for sub-AIS calcites, as the interior of the ice sheet – where we hypothesize calcite endmember waters originated – likely had much of the sedimentary cover stripped off during the ~35Myr glacial history (Graly et al., 2017 introduces this as a possibility for Greenland). This silicate dominated terrane would be analogous to the Canadian shield. Further, the $\delta^{13}\text{C}_{\text{DIC}}$ of PRR50489 calcites are ~3‰ more positive than their $\delta^{13}\text{C}_{\text{ORG}}$, which is consistent with the idea that the $\delta^{13}\text{C}_{\text{DIC}}$ of the water is equal to $\delta^{13}\text{C}_{\text{ORG}}$, and that the $\delta^{13}\text{C}_{\text{CARB}}$ was fractionated by ~3‰: an amount consistent with HCO_3^- - CaCO_3 fractionation. We have included text to describe this hypothesis in the manuscripts main text (also covered in lines 159-168).

We note that the requirement of full utilization of subglacial CO_2 would be difficult to achieve in Faure's hydrothermal spring source story. As hydrothermal waters would likely liberate high concentrations of carbon from the substrate and introduce them as CO_2 to the subglacial aqueous system. Rather than become fully utilized in silicate weathering, these high concentrations of carbon would drive silicate weathering until the solution had high alkalinity, at which point the dominant carbon species would be HCO_3^- , and excess CO_2 would fractionate by approximately 8‰. Instead, microbial oxidation of organic matter liberates a small amount of CO_2 , that is fully utilized during silicate weathering and produced calcite with $\delta^{13}\text{C}_{\text{DIC}}$ only 3‰ higher than $\delta^{13}\text{C}_{\text{ORG}}$.

I could not find how reference (35) supports that PRR50489 $\delta^{13}\text{C}$ values derived from carbon sourced from subglacial organic matter that is oxidized during microbial respiration.

In their supplementary materials, Yan, et al., 2019 describe high- CO_2 ice with $\delta^{13}\text{C}$ values of ca. -23‰ that can be attributed to microbial respiration at the base of the AIS. We've also added a citation for Montross, et al., 2014, which provide a similar result from debris-rich basal ice beneath Taylor glacier.

A $\delta^{13}\text{C}$ value of -23‰ suggests a very negative DIC or that the values are highly kinetically fractionated. This is a possibility for subglacial carbonates (cf Skidmore et al., 2004).

Given the -26‰ $\delta^{13}\text{C}_{\text{ORG}}$ within our calcites, along with the narrow range of ca. -23‰ compositions observed in PRR50489, we can rule out very negative DIC materials like methane. As mentioned previously, $\delta^{13}\text{C}_{\text{DIC}}$ of PRR50489 calcites are ~3‰ more positive than their $\delta^{13}\text{C}_{\text{ORG}}$, which is consistent with HCO_3^- - CaCO_3 fractionation at equilibrium in 0°C waters.

Such low value for the DIC species is actually consistent with Faure et al's (1998) hypothesis of hydrothermal fluids (even if the Author dismiss their hydrothermal fluids) of subglacial melt origin that percolated through the Beacon Supergroup in depth, became heated and stayed in contact with its lithofacies for a long time. I am not sure that the argument presented in this research paper rules out Faure et al's 1998 "hydrothermal" fluids.

To sum up our previous comments about the possibility of a hydrothermal input to PRR50489 and MA113: we feel that Faure's original interpretations of a hydrothermal origin of Elephant moraine precipitates, while consistent with what was known about subglacial processes in 1988, is no longer the best explanation for these samples given subsequent discoveries within Antarctic science that describe: 1) the widespread occurrence of subglacial waters; 2) the chemical characterization of these waters that match the compositions inferred from these samples. Several paradigm-shifting discoveries have been made in the two decades since Faure's publications that allow these samples to be viewed in a new light. Namely, the majority of the EAIS interior is at the pressure melting point and therefore there are long lived, isolated subglacial aqueous systems; cryoconcentration of these waters leads to subglacial brine formation; and microbial respiration drives relatively rapid chemical weathering of these rocks. Furthermore, we greatly expand on Faure's chemical characterization and find several lines of evidence that point to two, spatially distributed waters formed that the precipitates: an interior glacial meltwater and a brine. In these analyses, we do not find any isotopic evidence for hydrothermal fluids. We describe two samples, separated by 900 km, that both support this hypothesis. Unlike Faure's hypothesis, that would require hydrothermal waters in these distinct hydrologic catchments beneath the ice sheet, the hypothesis presented here calls on water types that have been observed in many different Antarctic sectors.

Neither is a good argument if considering Frisia et al. (2017) because their inference was that the calcites mostly precipitated from interconnected subglacial meltwaters. Closure of the system was documented by microsparite, which may be the equivalent of Opal in the EM and AM samples.

Despite the erroneous original citation, we agree with the idea that the calcites we observe have a formation mechanism tied to periodic interconnection and closure of subglacial meltwater. However, given our observation of millennial-scale flushing cycles that correlate with Southern Hemisphere climate, we call on a climate-ice sheet teleconnection to explain the flushing. We also do not observe any evidence for hydrothermal inputs to our subglacial system.

There is an alternative: methane oxidation during the precipitation of PRR 50489 calcite.

We've addressed this in previous comments.

The MA 113 DIC is 5‰ more positive than PRR 50489, which suggests more variable organic matter oxidation pathways. This should be an aspect to account for in the discussion that may shed light on the provenance of formation waters in the two study sites.

Based on our mixing models, the $\delta^{13}C_{DIC}$ of MA113 is more enriched than PRR50489 by ~3‰. We interpret this difference either as slightly different $\delta^{13}C$ of the organic matter at Mount Achernar or that there is a small amount of calcite in the bedrock that MAM waters contacted. Based on the overall similarities between chemical characteristics of MA113 and PRR50489 we do not expect different oxidation pathways at the two sites.

I suppose that Opal is CT. Then, it is most likely that it is a transformation of Opal A. It becomes likely that the $\delta^{18}O$ signal of opal could reflect kinetics related to such transformation. Structural changes in the silica framework may impart changes in the O isotope composition as observed in marine sediments consisting of opaline silica. During Si-O-Si condensation there may be an enrichment in the heavy isotope. Hence, the suggestion that the mixing model should take kinetics and diagenesis into some “form” of account.

Given that we have data showing the opal is opal-A we do not see any indication that the stable isotope values we present are not primary signals. We have made note of this in the manuscript text (line 114-117).

Faure et al. (1988) pointed out that the elevated $^{87}Sr/^{86}Sr$ values of the EM calcites are to be ascribed to contact of fluids with “old Rb-bearing rocks of the substrate”. Little is known about the MA samples, unless the bedrock was also Beacon supergroup. Now, the assumption that there is no significant strontium isotopic fractionation for calcite is correct, as it depends on the precipitation rate, which is likely to be slow (cf. Bohm et al., 2012). The spread of the $^{87}Sr/^{86}Sr$ values of calcite and opal is, in effect, consistent with various contribution from the Beacon Supergroup. But this assumption is based on the fact that there is no kinetic fractionation associated with the rate of Opal precipitation, the transformation Opal A-Opal CT and the fact itself that opal formation starts with the polymerization of a colloidal suspension. This is a complex nucleation pathway. I was wondering if there are studies that relate Sr uptake in opal and its fractionation in non-crystalline phases of SiO_2 . Most of the knowledge we have is on the crystalline form of minerals. Thus, the potential of transformations should be given insight when explaining the rationale of the model...maybe simply stating that diagenetic transformations were not taken into account but one should be aware that the kinetic processes associated with Ostwald's step rule may influence TE uptake and isotope fractionation.

To our knowledge there has not been any studies devoted to the Sr isotope fractionation in opal. However, the total variation of $\delta^{88}Sr$ in all terrestrial materials that have been observed amounts to 1.1‰ relative to the standard SRM987 (Teng, et al., 2017). Given that $\delta^{87}Sr$ has half of the mass difference, this would result in a change of 0.55‰ for $^{87}Sr/^{86}Sr$ values. Because we measure a radiogenic fractionation, the variation in $^{87}Sr/^{86}Sr$ in endmember waters both for of PRR50489 (~30‰) and MA113 (~2‰) is too large to be explained by this potential Sr isotope fractionation.

Line 156-81: Chlorine . The issue here is that chlorine is difficult to accommodate in the lattice of calcite. Cl is a monovalent anion, similar to fluorine. Thus, the two possible substitution mechanisms for Cl in calcite is replacing one oxygen atom within the CO₃²⁻ group to form a (CO₂Cl)⁻ group or substitution involving replacement of the CO₃ group by two Cl⁻ ions to form a CaCl₂ defect. The observed uptake of Cl and F decreases from vaterite to aragonite to calcite. Which means that Cl would be accommodated preferentially in a precursor phase in an Ostwald Step Rule series. I am not an expert in the SiO₂ system, but it appears that the hydrated forms of SiO₂, which may be a precursor phase according to Ostwald Step Rule, uptake Cl. Hence, opal may incorporate more Cl than calcite not because the water has a different provenance, but because opal is hydrated. I suggest this depositional pathway needs to be discussed in the framework of the significance of the PREEQC results. Crystallization pathways and kinetics associated with them are important in palaeo-science.

We make the inference of a chlorine-rich brine based on several chemical factors. Including the idea that the opal endmember must be low in carbon (so as to not precipitate abundant carbon), must have high $\delta^{234}\text{U}$ indicative of long-term water rock interaction, and must have $\delta^{13}\text{C}$ sourced from dissolution of silicates (i.e. $\sim -5\%$). We describe a Cl-rich brine as a likely candidate for the opal endmember because it matches these conditions, has been found in subglacial settings throughout Antarctica, and our PHREEQC models show that cryoconcentration of such a brine could lead to monomineralic opal precipitation.

We agree with the notion that Cl substitution and Ostwald Step Rule may affect the uptake of Cl in our samples. This can specifically draw concerns regarding Extended Data Figure 6 showing that Cl is abundant in opal layers relative to calcite layers. We originally included these data as secondary evidence for a Cl-rich brine, and do not view it as central to our argument for a brine endmember. As such we have removed the text describing Cl concentration within opal layers out of the main body of the manuscript.

In this framework, I believe that it is necessary to cite what the mineralogy of the subglacial carbonates found near Carey Station is ...I found it is aragonite. Not just provide a reference, please. Interestingly, Goodwin (1988) reports that “chemical and isotopic analysis of the jokulhlaup water determined its origin as “basal melt water which had been derived from the melting of ice from before the last glacial maximum” and that gained their chemical properties from the interaction with fine grained glacial deposits (which are very reactive). Then, if the mineralogy is aragonite, its formation requires different chemical parameters than those that form calcite. Then, the use of the jokulhlaup events chemistry as an end member for the mixing model may not reflect the actual chemistry of the subglacial meltwaters on the Ross Sea side of the Transantarctic Mountains, which Faure et al. (1998) suggest are influenced by interaction with the Beacon Supergroup. The Geology is rather different and it counts. This is the reason why some subglacial precipitates may show different minerals with diverse trace element concentrations and isotope ratios. The nature of sediment/bedrock in the Law Dome region, the bedrock in Boggs Valley are not the Beacon Supergroup. Therefore, the idea that formation of subglacial calcite in Antarctica may be related to the same large-scale climate needs, at first, to be tested by contextualizing them in their geologic/geomorphologic setting, with potential fluid composition

that evolved from the bedrock. The Law Dome subglacial fluids cannot be taken as end members for the EM and AM precipitates, simply because the minerals that would have been dissolved are very different and because coal in the Beacon Supergroup complicates the geochemical evolution of fluids that go through it (and of the calcites) . Otherwise, the deposits would have similar geochemistry and mineralogy, but they do not.

We've added mention of the Casey station mineralogy to the main text (line 201orbi).

We've also added descriptions of mineralogy for both sampling locations (lines 96 and 100). We note that both locations have very similar mineralogy dominated by Beacon Supergroup and Ferrar dolerite.

The requirements for opal and calcite precipitation: concentration of fluids that have experienced silicate weathering, and oxidation of organic matter respectively, are possible in a range of sub-AIS environments and geologic settings. Indeed, subglacial carbonate precipitates have been found across the continent in a range of water compositions. We therefore respectfully disagree that geologic context need be considered when assessing our samples as climate archives.

The two samples have both contacted dominantly Beacon supergroup and Ferrar dolerite. Although not presented in this manuscript, we have of evidence of detrital coal within MA113. The mineralogy and geochemistry of the two samples presented here, including $\delta^{13}C$, $\delta^{18}O$, and Sr, are relatively similar.

For example, the evidence from Boggs Valley is that the calcite crusts formed from subglacial meltwater ... not hydrothermal as stated in the present submission... during the Antarctica LGM, as precise ages suggest (see also the latest LGM synchronization between NH and SH by Svensson et al., 2020). The Frisia et al (2017) paper reports that pebbles cemented by calcite may be the result of catastrophic subglacial discharges into the valley from a lake that formed through the influence of volcanism between ca 24 and 22 ky, which coincide with a warm phase recorded by EDML and WDC (Svensson et al., 2020). With the exception of conglomerate-like "layer", most of their calcite seems to have grown in a cold period. Please amend the citations (Faure et al for the hydrothermal and Frisia et al for the basal ice melting) and put them in the correct perspective.

We have made the suggested edit.

Minor points:

I would like to point out that the map shown in sample in Fig 1b Extended dataset should have been carried out accounting for the dislocation in the middle of the sample. The direction of growth of the crust changes, so, the top part was analysed along the growth direction, the lower half is at an angle. What is the relationship with the ages of this transect?

We have more clearly labelled where the transect was taken in this figure and have added text to the figure caption to explain this.

The topmost part of MA 133 is different from the part below and appears to have some microsparite associated with calcite and opal. Given the complexity of the stratigraphy, and deformations, it is recommended to plot onto Fig 1 Extended Data where the dating samples were taken. And it would also be commendable to plot the Ca and Si W% for both in terms of distance from top. The plot vs ages in Fig. 1 of the main text is the interpretation. The raw dataset should be plotted with the microstratigraphy to allow a direct comparison with the petrographic features of the specimens.

We have added where the sample dates were taken to figures 2 and 3 and have added full-sample Ca versus z plots to figures S1 and S2.

Synthesis

I am aware that the only way to solve some issues would be to have reproducible data from at least a second sample of the same age in the EM or MA, so that correlations of stratigraphy and chemistry can be operated. But I am also aware that this may not be possible due to the fragmented nature of moraine specimens.

On the basis of one good sample it is difficult to prove the hypothesis, but I hope I have provided additional tools that may support it as well as some food for thought.

Reviewer #2 (Remarks to the Author):

Piccione et al. investigate a novel paleoclimate archive, a chemical precipitate, over two rather short periods (<100 kyr) during MIS 3 and 7/6, respectively. They argue that the calcium-rich endmember with low d18O values indicates a meltwater source in the interior of the EAIS, while the opal-rich end member with high d18O values point to meltwaters from a rather marginal ice position. The authors conclude that calcite precipitated in subglacial meltwater during AIM warm periods. They further infer a distinct alteration of opal- and calcite-rich precipitates related to millennial-scale changes, with calcite precipitating during the warmer AIM periods.

I find this an interesting and thoroughly carried out study. The main conclusion, i.e., AIM warm phases show higher delivery of meltwater from the interior of the ice sheet, is intriguing because it contradicts the concept that during a retreat ice melt is primarily enhanced at the edges of an ice sheet. I also find it intriguing that the chemical precipitate exhibits a switch between opal-rich and calcite-rich layers pointing to a threshold behavior in the response, typical for ice sheets, rather than a linear response to a smooth forcing. A third important implication is that the millennial-scale drivers seem to be sufficient to switch the meltwater system, i.e. there is no underlying orbital-scale change required. Even a glacial termination – which is not preserved in the record – could not do more than switching the chemical precipitate. In light of this, I believe this study could be of wide interest.

We thank Dr. Weber for the insightful review and appreciate the lending of his expertise for the benefit of our manuscript. The implications pointed out in this review have added

to our discussion, and his suggested edits have helped us prepare this revised version of the manuscript. We respond to each comment below:

As far as I can see the investigation is based on sound methods and results, although I need to emphasize that I am no expert in geochemical or ice-sheet modeling. But the chemical analysis, age dating and age model development seem sound from my point of view. Also, the authors have a clear understanding of how to put their results into a paleoclimatic context. Therefore, it is hard for me to point out any major weaknesses or flaws.

However, I am not expert enough to judge if their main interpretation of precipitate formation is well enough supported, i.e., oscillations between freezing of local brines from proximal sources during cold periods, and influx of far field EAIS meltwaters during warm periods.

According to the results of their ice-sheet modeling, millennial-scale changes are the main drivers of ice elevation changes of several hundreds of meters. This is interesting and in line with independent findings we made based on isotope studies in the Weddell Sea Embayment. It would also imply, as stated by the authors, that the Ross Sea behaved in a similar manner, a fact that has previously been questioned given, for instance, the later response of the WAIS to millennial-scale changes of the last deglaciation.

Even though the time scale is short and the records display primarily millennial-scale changes, I still recommend to display at least an insolation curve (I recommend Southern Hemisphere summer insolation for Dec-Feb at $\sim 75^{\circ}\text{S}$) for PRR50489 in Fig. 1, and discuss the potential impact of orbital forcing on the record. I do see a relation to the model, for instance, in Fig. S1, to both insolation and obliquity. Therefore, simply to avoid being forced to change the interpretation in case a longer precipitate record will be retrieved showing distinct orbital responses, I recommend including the discussion on orbital (ice-age) influence.

We have added the Southern Hemisphere summer insolation for Dec-Feb at $\sim 75^{\circ}\text{S}$ insolation curve to Figs. 2 and 3.

We agree that orbital forcing should be discussed. We've made note of this on line 271.

On that notion, it would be beneficial if all displays against time contain the MIS stages (as in Fig. 1).

We have added MIS stages to all figures with displays against time.

Comments on figures and manuscript arrangement

- I would prefer to see the map as Fig. 1 and data figures after that. It just makes the content more accessible.

We have changed the order of the figures so that the map is Fig. 1.

- The paper has clearly been written in the short format for, e.g., Nature. However, Nature Com allows for much more content in the main paper. Therefore, I recommend a major re-structuring

and re-organization of the paper. Right now there are very extensive supplements for both text/methods and figures, which is confusing and this cluttering is not helping to make a convincing case. The specific changes are up to the authors, however. Subsequently, the supplement should be reduced to one file containing necessary methods and figures that do not fit in the main paper.

We agree with this assessment and have made the suggested restructuring, which we outline below.

As for the additional figures in the main paper I recommend:

- Extended Data Fig. 9 could become the final figure of the main paper.

We have made this suggested change.

- Since this is such an unusual climate archive, it would also be beneficial to have Extended Data Fig. 1 – showing the actual precipitates – in the main manuscript.

The previous Extended Data Figure 1 is now incorporated into the main manuscript within figures 2 and 3.

- Extended Data Fig. 2 contains the age model with the Bayesian model, which would also be appropriate for the main manuscript.

Given space constraints, we've had to keep the Bayesian age-depth models in the supplementary materials.

- Extended Data Figs. 3 and 4 can also be displayed in the main manuscript, although they would need to be polished to match the style of Fig. 3.

We have added what was previously Extended Data Fig. 3 as Fig. 5 of the main manuscript.

We are no longer including the results from Extended Data Fig. 4 in the manuscript. We have replaced this with a thresholding experiment.

As for the supplementary methods I recommend to move to the main paper:

- The chapter 'Subglacial precipitate source areas'. This is clearly a discussion that fits the main paper.

Due to constraints on the maximum text length, we are not able to include the whole discussion of source area in the main text. We've added a short discussion of source area to the results section (lines 93-105) and refer to the supplement for a more detailed description.

- The chapter 'Reduced complexity ice sheet model' is right now a mixture of methods and interpretation. This chapter would best be split into the method itself, which should stay in the

supplement, and the interpretation, which should move to the main paper. I am not sure about the two associated figures but perhaps Fig. S1 could go to the main text as well (with some orbital curves –see above, and discussion thereof).

We have added the text describing the reduced complexity model of ice sheet thermodynamics to the results section “Millennial-Scale Ice sheet Variability (text added to each of the 5 paragraphs inn that section). Due to space constraints, we have left the two associated figures in the supplementary materials.

More specific comments by line:

- Lines 234-237

If you intend to elaborate on potential leads or lags, you need to discuss your dating uncertainties (the U/Th dates) with the uncertainties of other, Southern Ocean climate indicators. This will be tricky and as you state, beyond your abilities. I would simply mention that the linear relationship implies synchronicity (within dating uncertainties) although you can’t quantify leads or lags.

We have made this suggested edit (now lines 285-288).

- lines 280-282

What are the implications? Be specific.

We have reworded this sentence for clarity (now lines 338-341).

- line 349

exception rather than acceptance?

We’ve fixed this typo.

I would be happy to conduct a second review on a structurally revised manuscript.

Michael Weber

Reviewer #3 (Remarks to the Author):

Piccione and colleagues present two subglacial precipitate samples from the EAIS, close to the Transantarctic Mountains. These samples show variations of Ca concentrations at millennial scale variability for two time periods; from 230-150 kyr BP (PRR50489) and 54-42 kyr BP (MA113). These variations coincide with Antarctic ice cores δD signal, where low Ca concentrations correspond to antarctic cold periods and high Ca concentrations to warm Antarctic Isotope Maximum (AIM) peaks. Rich Ca concentrations occur as a consequence of enhanced basal melt production at the interior of the East Antarctic Ice Sheet (EAIS) during warm periods. The novelty of their study is that they claim that these variations in the hydrological cycle are explained through ocean warming, which produces grounding-line migrations. These migrations produce changes in the surface slope affecting ultimately the subglacial heat budget at the EAIS. If the measured signal can be directly related to changes in

the oceanic temperature, then this would confirm the existence of the bipolar-seesaw mechanism caused by changes in the AMOC. Such a finding is well suited for the scope of Nature Communications.

Overall, the manuscript is well written and easy to read for someone as me who is not familiar with subglacial precipitates. Since I cannot give an opinion on the geochemical analysis and the employed dating methods I have only one general comment regarding the mechanism that drives this signal and two technical questions.

General comment:

Atmospheric contribution

The key message of this manuscript relies on the basis that the origin of the signal comes from oceanic variations, however, couldn't this millennial variation be explained through atmospheric warming? It is stated (L292) that such an atmospheric warming could not penetrate to the base of the ice sheet due to insulation effects, however, results from the Greenland Ice Sheet show that atmospheric warming changes ice dynamics, which increases basal friction and consequently basal melt production (Karlsson et al., 2021). A warmer atmosphere reduces the viscosity of ice sheets and enhances the ice flow, increasing frictional heat production. How can you ensure that the recorded Ca variations occur due to oceanic warming provoking grounding-line migrations and not a warmer atmosphere? Maybe you could do a sensitivity experiment with your ice-sheet model changing the value of "A" (Ice viscosity parameter; Eq. S2) following an Arrhenius law dependent on the temperature.

Thank you for this thoughtful idea. However, the mechanism for transferring atmospheric warming to the base of the ice sheet that is elegantly described in Karlsson et al. (2021) for the modern Greenland Ice Sheet is not applicable to the East Antarctic ice sheet even under the modern, warm interglacial climate. The specific mechanism through which atmospheric warming impacts ice dynamics in Karlsson et al. (2021) is the surface meltwater penetration to the bed. Under modern conditions this happens abundantly in the ablation zone of Greenland ice sheet, where summer temperatures are regularly above 0°C and surface melt rates reach meters per year. Hence, in Greenland, moulins and surface lake drainages can transfer large volumes of surface meltwater to the bed and influence ice sheet dynamics. However, our study regions have mean annual temperatures of -35 to -45°C and do not experience any surface melting. Neither did they experience any surface melting during the time periods covered by our samples. Therefore, the findings of Karlsson et al. (2021) do not apply here. Hence, we are justified in basing our analysis of ice sheet response to atmospheric forcing on the approach taken by MacAyeal in his 1993 paper on the binge-purge model for Heinrich events. As we point out in our analysis atmospheric warming on millennial timescales cannot impact ice sheet dynamics sufficiently fast and with sufficient magnitude to be responsible for our observations. In the absence of surface meltwater to the bed, atmospheric warming must penetrate close to the ice base to significantly influence ice flow because both sliding and motion through internal ice deformation are concentrated at / near the base (e.g., Clarke et al., 1977). As we point out in our supplemental materials this takes longer than a few / several thousand years when thermal diffusion through ice is

the dominant mechanism.

Technical comments:

Reduced complexity ice sheet model

Which ice sheet model are you using? Is it published? Maybe I understood it wrong, but you don't incorporate any ocean forcing, but rather grounding line migrations to mimic the Ca cycles. Do you have any idea of which oceanic warming you would need in order to produce such a grounding-line migration?

Our reduced-complexity model of ice sheet thermodynamics is described in detail in the supplemental materials and represents largely a modification of MacAyeal's binge-purge model (1993). It is correct that the model does not explicitly include ocean warming. To our knowledge, the sensitivity of the Antarctic ice sheet to ocean thermal forcing on various timescales is still a matter of vigorous scientific debate. The only modeling paper that has specifically looked at this issue on millennial timescales is Blasco et al. (2019), which we cite in our manuscript. A comparison of our findings with the model outputs published by Blasco et al. (2019) does indicate that the ice sheet would have to quite high sensitivity to ocean thermal forcing to explain our observations.

Ice core imprints

Your samples show periodic episodes of basal melt, is this also observable on ice cores close to the grounding line or ice cores close to the sampled regions?

Unfortunately, no such ice core records exist. Episodes of basal melting represent destructive events, akin to erosion of the ice column from below, and are not recorded directly in ice cores. They could be recorded indirectly as periods of ice elevation drop during millennial-scales warm climate phases whose presence one could infer from the records contained in ice cores, just as it has been done for ice elevation changes on glacial-interglacial timescales (e.g., Buizert et al., 2021). However, doing this will be more difficult on millennial timescales over which these elevation changes are smaller than during glacial-interglacial transitions. For instance, our own model output indicates that ice thickness changes on millennial timescales only by dozens of meters in the regions of sample precipitation. Once such ice thickness perturbations propagate to the ice divide regions where ice cores are typically obtained, the magnitude of these perturbations will be even smaller.

References:

Karlsson, N.B., Solgaard, A.M., Mankoff, K.D., Gillet-Chaulet, F., MacGregor, J.A., Box, J.E., Citterio, M., Colgan, W.T., Larsen, S.H., Kjeldsen, K.K. and Korsgaard, N.J., 2021. A first constraint on basal melt-water production of the Greenland ice sheet. *Nature Communications*, 12(1), pp.1-10.

REVIEWERS' COMMENTS

Reviewer #1 (Remarks to the Author):

The Authors have carefully taken into consideration all the comments by the Reviewers.

The manuscript has been considerably improved and is providing strong support for the hypotheses formulated.

The supplementary material has also been completed by important information for future referencing.

The manuscript is very well written and pleasant to read and, I am sure, advances our understanding of the role of Antarctic processes in global changes, as well as on Antarctic ice sheet responses to millennial climate changes.

I am wholly satisfied with the argumentation provided in the Rebuttal to Reviewer 1 and wishes to suggest publication of the manuscript likely in its present form.

I have a just few minor suggestions, which may be considered (or not) by accounting for the word limits and sentence construction.

Line 62: I am aware it is not relevant, but many palaeo-climate researchers refer to the southern shift of the ITCZ as a response to NH cooling. Thus, I suggest to change the sentence as follows (although I agree that reading may be less smooth): ...north during SH/NH cold/warm periods and south during SH/NH warm/cold periods.

Line 64: The statement:...."dampen millennial-scale climate variability" is unclear. Is it referred to the amplitude of changes?

Line 66.. truth or true?

Line 100: larger or longer?

Line 107 cyclic changes

Line 112: may I suggest replacing "fibrous" with "acicular"? "Fibrous" is commonly used to refer to calcite crystals with pliable quality, such as in moonmilk and tufa. Given the similarities with Boggs Valley and with some speleothem fabrics, the term columnar calcite (for the bladed fabric) and

elongated columnar (or acicular) for the acicular fabrics would facilitate comparison with other continental fabrics (cf. Frisia et al., 2017; Frisia, 2015).

Also, in relation to opal, perhaps it is: nucleation in the water column followed by particle settling.

Line 114: may I suggest using “diagenetic transformations” rather than alterations? It would convey the idea of Ostwald Step Rule.

Line 117. Please omit “unaltered” opal A. One does not really know what the actual crystallization process might have been.

Line 221. There is a change in font....(geochemical)

Line 249: insert a space between rates and (e.g. interglacials).

Line 308. Perhaps ...each consisting of..., rather than each made up of....

Line 370 ..”the subglacial precipitate record”...

Reviewer #2 (Remarks to the Author):

The author's have responded to my comments satisfactory. The issues I had (inclusion of orbital forcing and the re-structuring of the manuscript) have been addressed. The ms is now much better accessible. The supplement has been reduced in size and much content has been transferred to the main paper, especially those aspects that included discussion.

However, I am still confused that there is a file with the supplementary text containing Figures S1 and S2 and a separate file for the supplementary figures containing Figs. S1-S8. Yet, Figs. S1 and S2 in both documents are obviously not similar. This discrepancy should be addressed.

Reviewer #3 (Remarks to the Author):

I do not have any further comments since my main concerns were answered.

Reviewer #1 (Remarks to the Author):

The Authors have carefully taken into consideration all the comments by the Reviewers.

The manuscript has been considerably improved and is providing strong support for the hypotheses formulated.

The supplementary material has also been completed by important information for future referencing.

The manuscript is very well written and pleasant to read and, I am sure, advances our understanding of the role of Antarctic processes in global changes, as well as on Antarctic ice sheet responses to millennial climate changes.

I am wholly satisfied with the argumentation provided in the Rebuttal to Reviewer 1 and wishes to suggest publication of the manuscript likely in its present form.

I have a just few minor suggestions, which may be considered (or not) by accounting for the word limits and sentence construction.

Line 62: I am aware it is not relevant, but many palaeo-climate researchers refer to the southern shift of the ITCZ as a response to NH cooling. Thus, I suggest to change the sentence as follows (although I agree that reading may be less smooth): ...north during SH/NH cold/warm periods and south during SH/NH warm/cold periods.

We've revised this sentence to now describe ITCZ shifts with respect to the AMOC driving mechanism rather than the hemispheric temperature. This, we feel, mitigates the possible interpretation that we are referring to the ITCZ shift as a function of SH temperature. We've also added the corresponding hemispheric temperature in parenthesis (i.e., NH/SH warm/cold periods and vice versa).

Line 64: The statement:...."dampen millennial-scale climate variability" is unclear. Is it referred to the amplitude of changes?

We have changed this statement to ... "dampen the amplitude of millennial-scale climate variability".

Line 66.. truth or true?

We've replaced "ground truth" with "support".

Line 100: larger or longer?

We've changed "larger" to "longer".

Line 107 cyclic changes

We've made this suggested edit.

Line 112: may I suggest replacing “fibrous” with “acicular”? “Fibrous” is commonly used to refer to calcite crystals with pliable quality, such as in moonmilk and tufa. Given the similarities with Boggs Valley and with some speleothem fabrics, the term columnar calcite (for the bladed fabric) and elongated columnar (or acicular) for the acicular fabrics would facilitate comparison with other continental fabrics (cf. Frisia et al., 2017; Frisia, 2015).

We've made this suggested edit.

Also, in relation to opal, perhaps it is: nucleation in the water column followed by particle settling.

We've made this suggested edit.

Line 114: may I suggest using “diagenetic transformations” rather than alterations? It would convey the idea of Ostwald Step Rule.

We've made this suggested edit.

Line 117. Please omit “unaltered” opal A. One does not really know what the actual crystallization process might have been.

We've made this suggested edit.

Line 221. There is a change in font....(geochemical)

We've fixed the change in font.

Line 249: insert a space between rates and (e.g. interglacials).

We've made this suggested edit.

Line 308. Perhaps ...each consisting of..., rather than each made up of....

We've made this suggested edit.

Line 370 ..”the subglacial precipitate record”...

We've made this suggested edit.

Reviewer #2 (Remarks to the Author):

The author's have responded to my comments satisfactory. The issues I had (inclusion of orbital forcing and the re-structuring of the manuscript) have been addressed. The ms is now much

better accessible. The supplement has been reduced in size and much content has been transferred to the main paper, especially those aspects that included discussion.

However, I am still confused that there is a file with the supplementary text containing Figures S1 and S2 and a separate file for the supplementary figures containing Figs. S1-S8. Yet, Figs. S1 and S2 in both documents are obviously not similar. This discrepancy should be addressed.

We have fixed this discrepancy by adding the two supplementary figures from the reduced complexity model of ice sheet thermodynamics as figures S9 and S10.

Reviewer #3 (Remarks to the Author):

I do not have any further comments since my main concerns were answered.